# Continuous ammonia electrosynthesis using physically interlocked bipolar membrane at 1000 mA cm⁻²

Ziang Xu[1], Lei Wan[1], Yiwen Liao[1], Maobin Pang[1], Qin Xu[1], Peican Wang[1] & Baoguo Wang ●[1] ✉

Electrosynthesis of ammonia from nitrate reduction receives extensive attention recently for its relatively mild conditions and clean energy requirements, while most existed electrochemical strategies can only deliver a low yield rate and short duration for the lack of stable ion exchange membranes at high current density. Here, a bipolar membrane nitrate reduction process is proposed to achieve ionic balance, and increasing water dissociation sites is delivered by constructing a three-dimensional physically interlocked interface for the bipolar membrane. This design simultaneously boosts ionic transfer and interfacial stability compared to traditional ones, successfully reducing transmembrane voltage to 1.13 V at up to current density of 1000 mA cm⁻². By combining a Co three-dimensional nanoarray cathode designed for large current and low concentration utilizations, a continuous and high yield bipolar membrane reactor for NH₃ electrosynthesis realized a stable electrolysis at 1000 mA cm⁻² for over 100 h, Faradaic efficiency of 86.2% and maximum yield rate of 68.4 mg h⁻¹ cm⁻² with merely 2000 ppm NO₃⁻ alkaline electrolyte. These results show promising potential for artificial nitrogen cycling in the near future.

Ammonia (NH₃), as one kind of the most crucial and fundamental compounds used as fertilizer, chemicals as well as an important energy storage medium (108 kg H₂/m³ NH₃ at 20 °C and 8.6 bar) without carbon emission, is appealing for a total output of approximately 175 million tons worldwide annually with a market value of around USD 70 billion[1–4]. Haber–Bosch process, though providing human reality with needful NH₃ for over a century, is always called to be superseded by new-generation NH₃ synthesis technologies nowadays for its harsh production conditions that are far from normal temperature (400−500 °C) and pressure (>10 MPa) as well as inevitable CO₂ emissionss[5–7]. Alternatively, electrochemical route is describing a compelling way of NH₃ synthesis from widespread nitrogen sources[8,9].

Electrosynthesis of NH₃ from 8e⁻ nitrate reduction (8e⁻ NO₃⁻RR) received extensive attention recently for widespread reactant on both geography and concentration as contaminations[10,11]. In addition, industrial feasibility is also promised with high selectivity based on this route by avoiding activation of inert N≡N triple bond (941 kJ mol⁻¹) and ultralow N₂ solubility in aqueous solutions. Plenty of efforts have been made to improve catalytic activity and selectivity for 8e⁻ NO₃⁻RR to ammonia[12–15], e.g., by constructing Cu-incorporated organic crystalline[5] or Fe single atom on carbon[16], and already achieved nearly 100% faradaic efficiency under aqueous conditions. These developments of non-noble catalysts are urging NH₃ electrosynthesis in an alkaline system (beneficial to half reaction kinetics) to the final step before operating with a flow reactor as a continuous industrial process with high yield rather than in H-cells with short-term stability (see Supplementary Note 1 for detailed discussion). An ion exchange membrane is necessary for electrosynthesis devices for separating unsymmetrical electrolytes on both sides and preventing NH₃ from diffusing to the anode and being re-oxidized[17]. Nevertheless, a monopolar ion exchange membrane, either an anion exchange membrane or a cation exchange membrane, can hardly fulfil the process and will lead

[1]Department of Chemical Engineering, Tsinghua University, Beijing, China. ✉e-mail: bgwang@tsinghua.edu.cn

to severe ionic (alkali metal ions or $NO_3^-$) crossover, as shown in Fig. 1. A bipolar membrane (BM) composed of an anion exchange layer and a cation exchange layer, by means of which ions with opposite charge can be repulsed by Donnan potential and $H^+$/$OH^-$ can be produced by water dissociation (WD) at the interlayer[18–22] and move outwardly to constitute a circuit, provides a prospective answer to the dilemma theoretically (see detailed discussion in ii of Supplementary Note 1). Meanwhile, a BM reactor potentially solves the fundamental contradiction between the demand for hydrogen element for 8e$^-$ $NO_3^-$RR and the requirement of suppressing hydrogen evolution reaction (HER) at cathodic catalytic sites.

Although a bipolar membrane $NH_3$ electrosynthesis reactor affords the possibility to achieve a high yield rate and operation durability, existing commercially available (e.g., Neosepta BP1) or previously reported BM materials are mainly designed for application at low current density (<100 mA cm$^{-2}$), e.g. electrodialysis[23,24] or $CO_2$ electroreduction[25,26] processes. In reality, working at high current density will bring BMs with exponential challenging, mainly expressing in two aspects: (i) insufficient catalytic sites for WD reaction at the interface of anion exchange layer (AEL) and cation exchange layer (CEL); (ii) ballooning of AEL and CEL interface[27]. Conventional wisdom has provided rational designs for interfacial catalysts[28–31] or structure[32–34] for optimization of BMs under the discipline of protonation–deprotonation mechanism; nevertheless, while most efforts hardly take effect averting the predicament of both points simultaneously, especially for high current density. Incremental catalysts promote WD kinetics but enlarge the physical space between AEL and CEL, thus inevitably increase ionic transportation resistance and

the possibility of interfacial blistering. Hence, the architecture of a stable BM for continuous $NH_3$ electrosynthesis at high current density still remains a struggle.

Inspired by the traditional intelligence of Chinese carpentry, we developed a BM with the 3D physically interlocked interface, which enjoyed a "Mortise−Tenon Joint" structure (MBM). Patterned by synthetic CoNi hydroxide needle-like array templates, BM interlayers with embedded joints in sub-micrometer were fabricated and regulated, revealing superiorities in two aspects: (i) WD catalytic sites were maximized without expanding ionic transportation resistance and WD efficiency was boosted at high current density (1.13 V transmembrane voltage drop at 1000 mA cm$^{-2}$); (ii) blistering and separating of AEL and CEL were avoided with altered swelling directions and proliferated physical contact area, thus lifespan was effectively prolonged. By coupling with Co 3D nanoarray cathode that both catalytic sites and mass transfer boosted, continuous bipolar membrane $NH_3$ synthesis in flow reactor achieved at 1000 mA cm$^{-2}$ with Faradaic efficiency of over 86.2% and yield of 68.4 mg h$^{-1}$ cm$^{-2}$ using merely 2000 ppm $NO_3^-$ alkaline electrolytes. Moreover, a >100 h operation at 1000 mA cm$^{-2}$ also endorses the confidence of using MBM in high efficiency and yield rate $NH_3$ electrosynthesis technology from industrial effluents.

## Design philosophy and architecture of MBM

In general, the water dissociation (WD) performance of a bipolar membrane (BM) is determined by both WD reaction rate and ionic transportation at the interface between the anion exchange layer (AEL) and cation exchange layer (CEL) under reverse bias operation. Numerical simulation can help to figure out the rate-determine step of

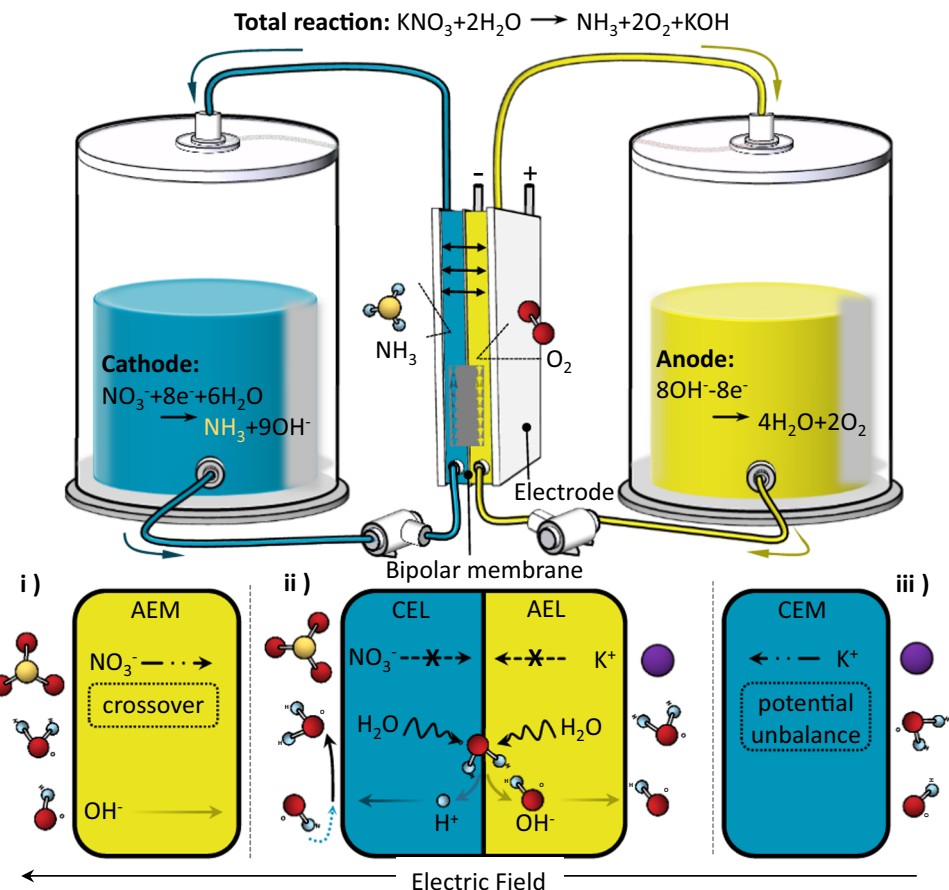

**Fig. 1 | Schematic illustration of continuous NH$_3$ electrosynthesis with a bipolar membrane reactor.** The flow directions of ions referred to in this process when adopting (i) anion exchange membrane or (ii) bipolar membrane or (iii) cation exchange membrane as the separator are shown, respectively.

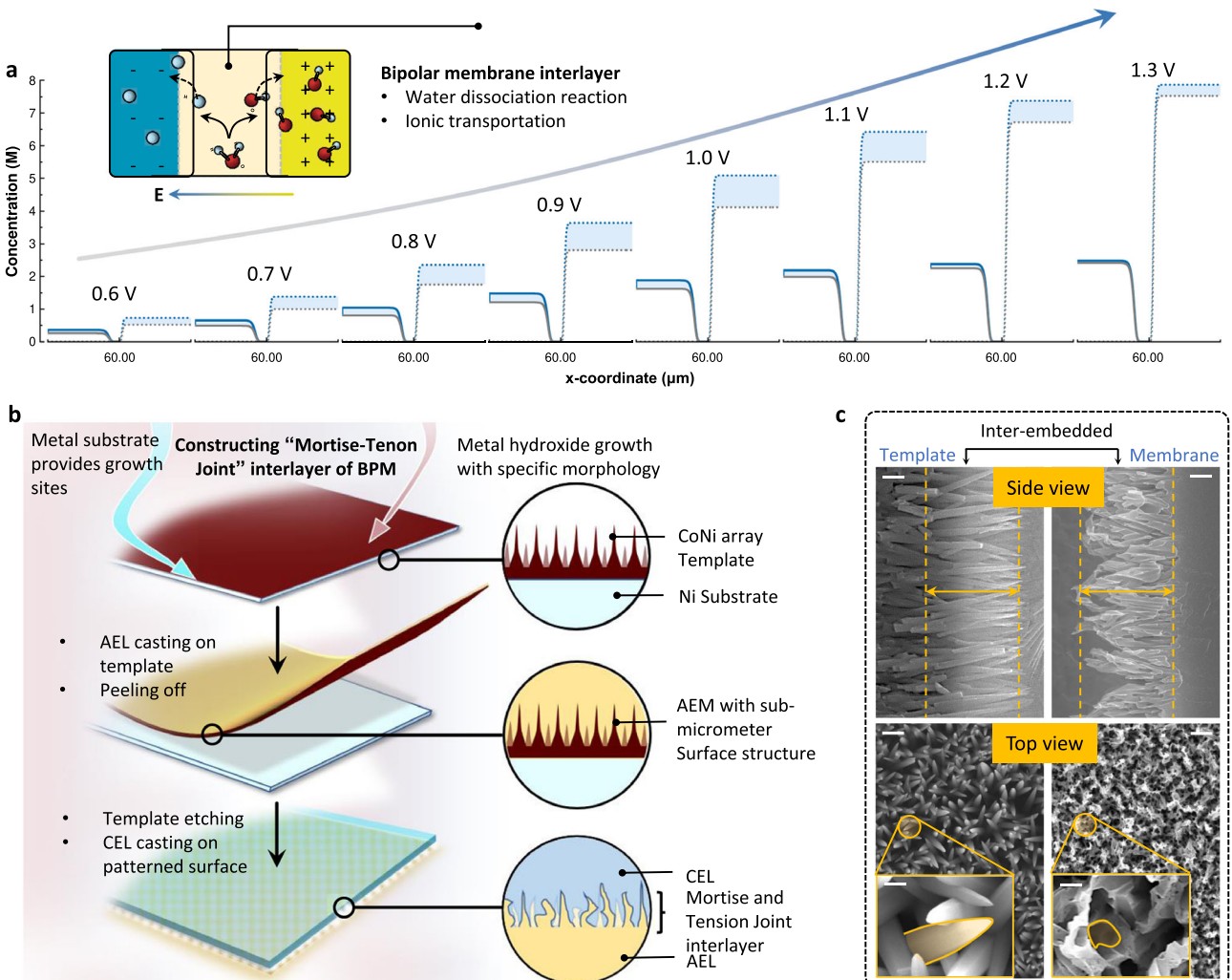

**Fig. 2 | Design philosophy of BM and illustration of MBM architecture process.**
**a** Concentration profile of H⁺ (solid) and OH⁻ (dot) near the interface of AEL and CEL with applied transmembrane voltage from 0.6 to 1.3 V based on numerical simulation via COMSOL software. The disparity of BM with (blue line) and without (black) enhanced interfacial kinetics is presented as a wathet lump. The x-coordinate of the interface is set to $x = 60$ μm and a 20 nm region of both layers is

presented here. Schematic of two steps of bipolar membrane WD process at interface is presented as inset. **b**, Schematic of multi-step MBM fabrication processes. **c**, Scanning electron microscope (SEM) image of CoNi hydroxide templates (left) and surface structure of template-transfer AELs (right) with scale bars of 1 μm. Inset shows a locally amplified structure of top view microstructure, with scale bars of 200 nm.

a bipolar membrane process and offer instructions for structural design. As shown in Fig. 2a (details can be found in Supplementary Note 2), the concentration of both H⁺ and OH⁻ dissociated from water molecules (close to the interface) can be boosted with the enhancement of WD rate constant and ionic diffusion rate in the voltage range from 0.6 to 1.3 V applied to BMs. This phenomenon demonstrates that the reaction and transportation steps of a BM interface can spur the overall WD kinetics coherently and optimizing either of both can only make a partial contribution (Supplementary Figs. 1–4, see Supplementary Note 2 for modeling details and further discussions). Hence, theoretical simulation emphasized the significance of constructing a BM structure for realizing synchronous promotion of both interfacial kinetics.

The traditional art of Chinese carpentry inspired us to construct a BM with three dimensional (3D) "mortise-tenon joint" structure interface (MBM) through a template-transfer strategy, aiming at interlocked AEL & CEL with surged contact sites. Quaternary ammonia poly (N-methyl-piperidine-co-p-terphenyl) (QPPT) and perfluorinated sulfonic acid were selected as AEL and CEL in MBM construction for their first-tier conductivity, chemical stability as well as proper swelling nature (Supplementary Figs. 5, 6). As shown in Fig. 2b, the template was firstly

prepared by self-growth of CoNi hydroxide on the metal substrate under hydrothermal method to obtain a needle-like microarray, and then QPPT ionomer solutions were cast on this template surface for AEL fabrication (see chemical structure in Supplementary Figs. 7, 8). After peeling off from the substrate and etching the embedded microarray, the AEL with a micro-patterned surface was obtained. Finally, the fabrication of MBM was completed by spray-coating of WD catalysts ink and perfluorinated sulfonic acid CEL on the patterned side of AEL, hereto "mortise-tenon joint" structures were achieved.

Physical parameters of "Mortise−tenon joint" structural interface could be precisely controlled by adjusting template morphology. As shown in Fig. 2c, the needles were dispersed uniformly with a length of around 2 μm, and the ratio of length to diameter was up to ~22.9. Such elaborate array architecture could form AEL-CEL interlocking layer with densely dispersed micropores. By overcoming the drawbacks in previous studies for 3D interlayers (electrospinning or photoetching template)[33,34], this approach realized MBM with highly ordered interlayer structure at sub-micro scale and conceivably offers MBM with not only abundant WD catalytic sites but also a strong combination of both layers for relieving blistering or delamination of AEL and CEL in practical applications.

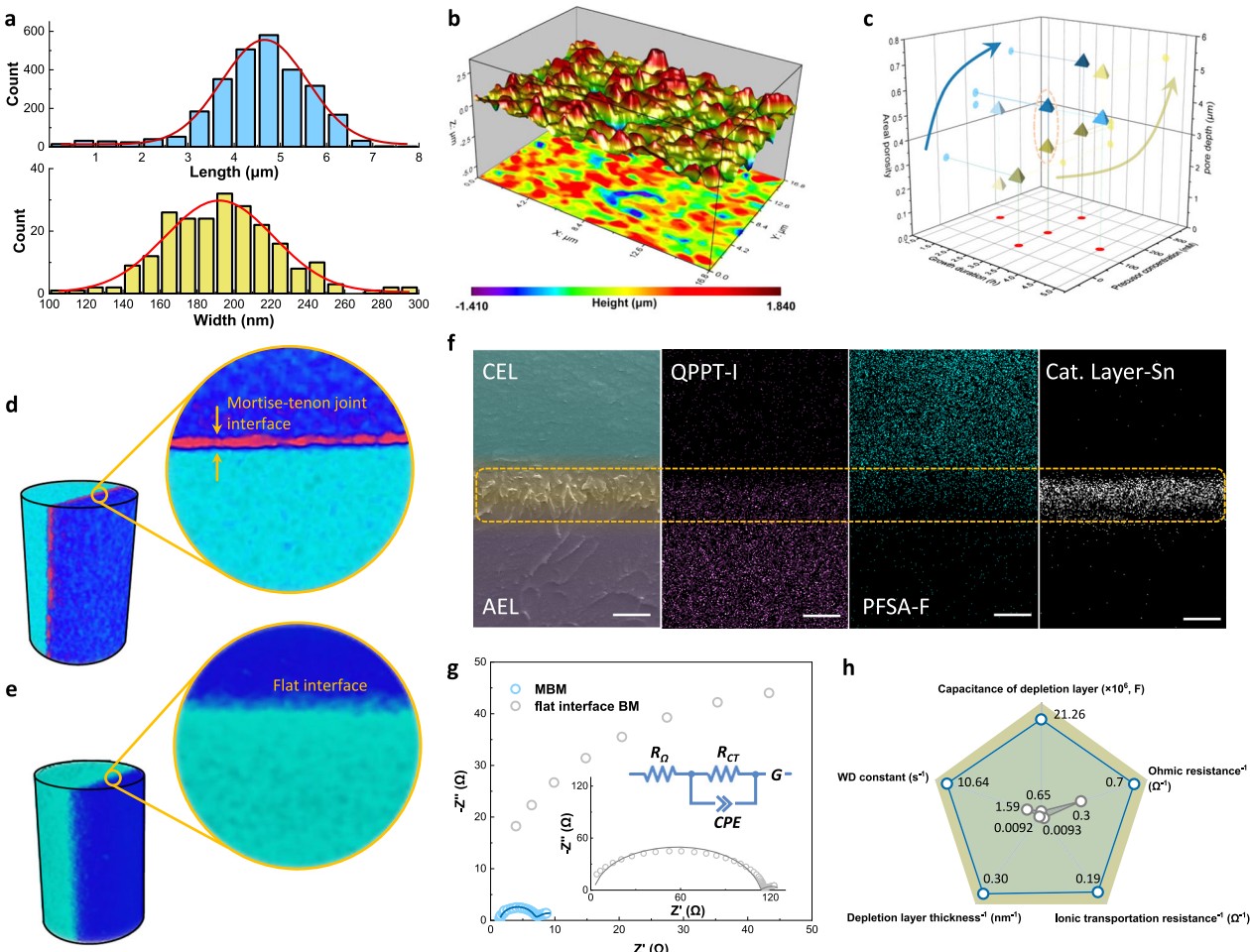

**Fig. 3 | Regulation and characterization of MBM interlayer structure and determination of WD kinetics. a** Distribution of heights (upper, wathet bar) and width (lower, yellow bar) of CoNi needle-like microarray as templates with their Gaussian profiles showing by red line plots. **b** Surface topography of template-transfer AEL re-plotted with lattice coordinates. **c** Independent regulations of AEL structural parameters on porosity and pore depth. Reconstruction of three-dimensional images and projection images of micro-computed tomography showing in **d** for MBM and **e** for flat interface BM obtained by in-situ micro-CT tomography with scale bars of 5 μm. **f** SEM image and elements distributions of MBM interlayer region with scale bars of 2 μm. The characteristic elements for the three layers are iodine (I), fluorine (F), and tin (Sn), respectively. **g** EIS Nyquist plots of MBM and flat interface BM at 5 mA cm⁻². Inset shows an equivalent circuit, where $R_\Omega$, $R_{CT}$, CPE, and $G$ represent Ohmic resistance, Ionic transportation resistance, constant phase element, and Gerischer element, respectively. **h** Fitting results of EIS measurements according to the equivalent circuit. Ohmic resistance, Depletion layer thickness, and Ionic transportation resistance were presented in reciprocal form.

## Confirmation of 3D structure and WD kinetics

The interfacial structure of MBM was estimated based on the uniformity and adjustability of physical morphology, and they potentially reveal a strong correlation with WD performance. As a typical case shown in Fig. 3a, the value of length and width for every needle centrally distributed around 4.67 μm and 192 nm, indicating an evenness of template by hydrothermal method (see Supplementary Note 3 for statistical method). The corresponding geometrical shape of patterned arrays could be easily transferred to the AEL surface, which was evidenced by its space lattice coordinates (Fig. 3b). By regulating growth durations or precursor concentrations of hydrothermal process (Supplementary Figs. 9–13), both the length and width of needle-like arrays were regulated in micron (average length: 1.28–8.41 μm) or sub-micron (average width: 92–261 nm) scale, thus a series of AELs with different areal porosity and pore depth can be achieved (Fig. 3c and Supplementary Figs. 14–17).

A 3D reconstruction of micro-CT was further employed to physically describe the MBM interface and compared it with a flat interface BM. As shown in Fig. 3d, e and Supplementary Figs. 18, 19, and Video I & II, 3D projection images revealed strong signals for X-ray transmitting

at any section planes of MBM, and these signals were obtained by abundant AEL–CEL contact area. On the contrary, negligible reflection signals were observed for flat interface BM interlayer with sharp contrast to MBM, ascribing to much less interfacial contact between two membrane layers. On the other hand, the profile of characteristic elements (I in AEL and F for CEL) on the interface delivered the same results, where the polymer of both layers interlocked each other with a depth of at least 2 μm, and catalytic particles scattered throughout this region (Fig. 3f). It can be found that the 3D mortise–tenon joint structure of MBM provides abundant contact sites of both membrane layers as pre-designed, which reveals entirely difference of interfacial morphology from the ones of traditional bipolar membranes.

In order to correlate the relationship among interfacial structure, WD reaction rate, and ionic transportation kinetics of BMs, electrochemical impedance spectroscopy (EIS) measurements were conducted (Supplementary Fig. 20)[30,35,36] and an equivalent circuit containing three serial parts was employed to outline the WD process (Fig. 3g). Five parameters can be derived by the fitting of Nyquist plots to make judgments on the corresponding interfacial behaviors (four independent ones except the depletion layer thickness derived from

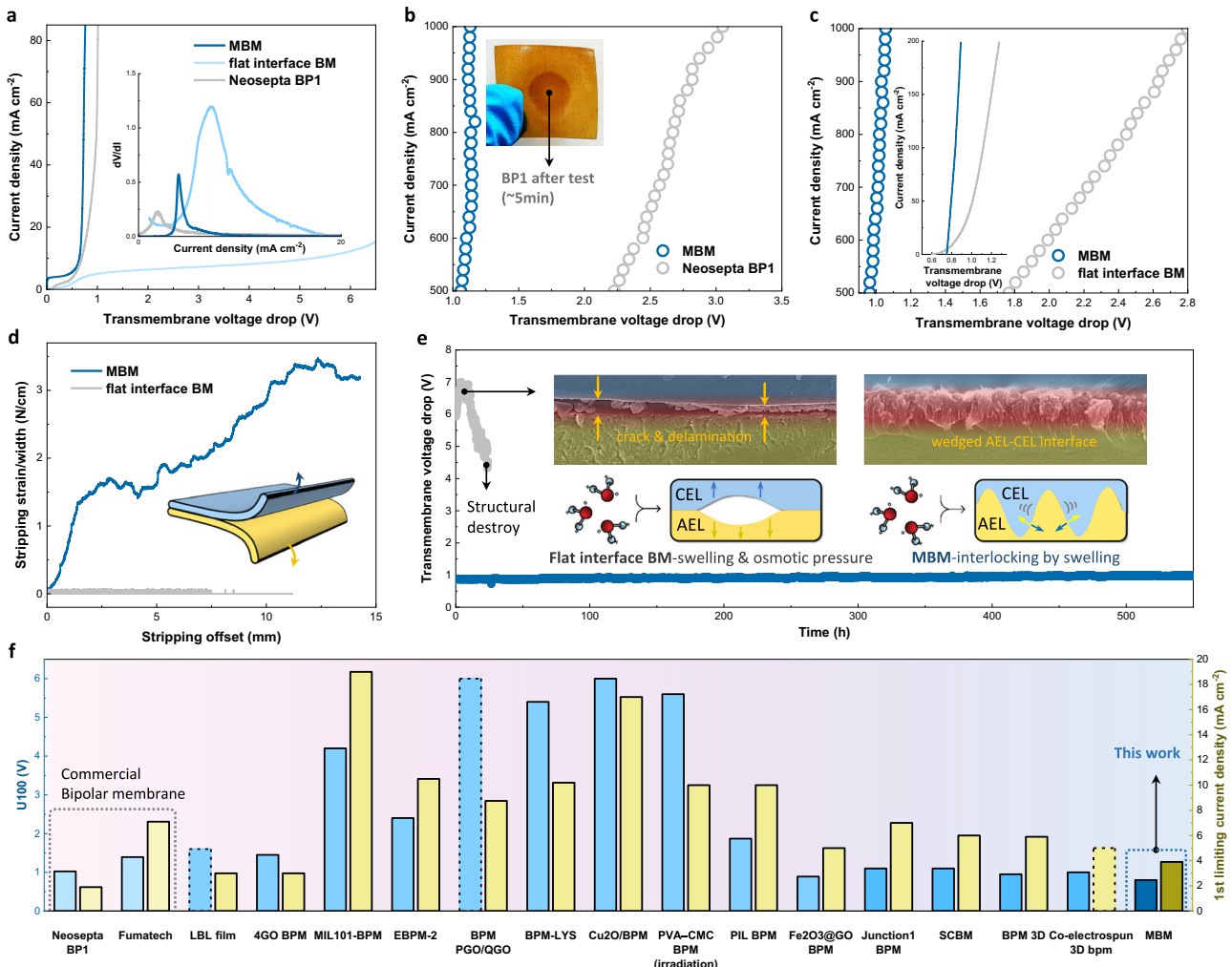

**Fig. 4 | WD performance characterization and stability of BMs.** Comparison of *I–V* curves for three BMs in **a** low current density and **b** high current density, measured in 0.5 M Na₂SO₄ aqueous solutions with a scan rate of 2 mA s⁻¹. Inset of **a** shows determination 1st limiting current density. Inset of **b** shows photography of BP1 image after one *I–V* cycle from 0 to 1000 mA cm⁻². **c** Comparison of *I–V* curves for BMs up to 1000 mA cm⁻² (inset is *I–V* curve in low current density) using unsymmetrical 1 M HCl/1 M KOH electrolytes with a scan rate of 2 mA s⁻¹. **d** Physical stripping strain of AEL–CEL of bipolar membranes in wet state versus offset

measured as shown in inset schematic. **e** Long-term WD stability measured in 0.5 M Na₂SO₄ at 300 mA cm⁻² over 550 h. The gray and mazarine plots represent the flat interface BM and MBM, respectively. Upper two insets show the microscopic structure of the AEL–CEL interface of the BMs; the lower two schematically show the mechanism of ballooning of flat interface BM (left) and that alleviated by MBM (right). **f** Comparison of electrochemical properties of BMs commercially available or reported recently (with advanced WD catalysts or structured interlayered design) based on dimensions of U100 and 1st limiting current density[18,23,30,32,35,39–47].

capacitance). As shown in Fig. 3h, although catalyst particles of the same species and similar loading were adopted for each membrane, the MBM revealed an enhanced WD constant (10.64 s⁻¹), about 6 folds higher than that of flat interface BM. These results can be attributed to the enlarged interfacial area between AEL and CEL, which brings bulky "effective" catalytic sites for WD reactions with the 3D interface. Furthermore, nearly 33 folds of depletion layer capacitance (21.26 μF) of MBM was observed, also indicating that a broadened physical interlayer produced more electrochemical active sites. The mortise–tenon joint provided an increased inner surface for catalytic sites and decreased the distance of both layers at a given loading; as a result, both depletion layer thickness (3.33 nm) and ionic transfer resistance (5.28 Ω cm²) were observably reduced compared to the flat interface BM, implying a lower energy barrier required for H⁺/OH⁻ to access membrane layers. In addition, although various interlayer geometrical parameters made distinct influences upon WD reaction and ion transport resistance, all these MBMs in our study revealed a boosted WD rate relative to the flat interface BMs (Supplementary Figs. 21, 22, find a detailed discussion in Supplementary Note 4). All these universal

promotion results of interfacial kinetics together confirm the surged sites and short mass transfer way brought by the mortise–tenon joint 3D interface.

## WD performance and stability of MBM
The mortise–tenon joint 3D interface anticipates MBM with an energy-efficient WD performance. For this purpose, *I–V* curves (transmembrane voltage trop vs. WD current) were collected in 0.5 M Na₂SO₄ as a comparison with flat interface BM and Commercial Neosepta BP1. As shown in Fig. 4a, the MBM far prevailed over flat interface BM at any current density and also revealed a preponderance of lower transmembrane voltage drop (0.8 V at 85 mA cm⁻²) than commercial BP1 (similarity can be found for other MBMs in Supplementary Fig. 23). The results indicate that less energy consumption was required to produce the same quantity of H⁺/OH⁻ compared to other BMs and can be concluded to a superior WD performance of MBM. Meanwhile, the lower 1st limiting current density (inset of Fig. 4a and Supplementary Fig. 24) confirms MBM with minor co-ion leakage (current density platform <5 mA cm⁻²)[37].

Both WD efficiency and selectivity of MBM prove mortise–tenon joint as a valid design for further application.

Bulky catalytic sites and lower ion transport resistance created by the 3D interface can further promote WD rate at high current density theoretically. For completely unmasking the feature, the WD rate up to 1000 mA cm$^{-2}$ was measured, and the results showed a distinct difference between MBMs and commercial BP1 (Fig. 4b and Supplementary Fig. 25). The transmembrane voltage drops of MBM stayed at 1.06 V at 500 mA cm$^{-2}$ and just increased into 1.13 V at 1000 mA cm$^{-2}$, presenting a 0.14 V/A cm$^{-2}$ slope of rise, while the commercial BP1 requested a 3.05 V to drive WD reaction and ionic motion of same rate with an increasement of 1.66 V/A cm$^{-2}$. As a matter of fact, the WD reaction can be boosted exponentially with increasing electric field, and the rate of interfacial ionic transportation as well as water diffusion play a bottleneck role in the high current region to determine overall WD performance[22,38]. Benefitting from the mortise–tenon joint with a large interfacial area, the distance between membrane layers was lowered guaranteed by sufficient WD sites, thus the transportation of both H$^+$ and OH$^-$ for approaching CEL and AEL was apparently strengthened. As a result, WD using MBMs consumed only power of 1.13 W cm$^{-2}$ at 1000 mA cm$^{-2}$, presenting an obvious energy-saving property relative to BP1 in ampere-level electrochemical devices (3.05 W cm$^{-2}$). The irreversible crimson of post-measured BP1 also set forth the defect of using traditional bipolar membranes in large current applications (inset of Fig. 4b). As a potential candidate for the emerging electrochemical devices with electrolytes far from neutral (e.g. alkaline NH$_3$ electrosynthesis discussed below) or even unsymmetrical (e.g., 1 M H$^+$/OH$^-$), MBM also reveals the superiority of energy efficiency for WD compared to commercial BP1 as shown in Fig. 4c and Supplementary Fig. 26.

The mortise–tenon joint promises the stability of MBM for long-term WD operation by potentially enhancing the binding force between AEL and CEL as well. Normally, inevitable physical ballooning at the interface will lead to a burgeoning energy consumption of BM electrochemical devices[39]. To mimic the operation status, we set up a measurement by imposing a stripping strain on each layer to evaluate the capacity of BMs that can confront blistering during operation (inset of Fig. 4d). As shown in Fig. 4d, MBM revealed several orders of magnitude higher stripping strains compared to flat interface BM, indicating that interfacial tightness was enhanced by fabricating a mortise–tenon joint interface. Ascribing to this merit, an over 550-h stable WD operation was realized at a current density as high as 300 mA cm$^{-2}$ in stark contrast to the structural destruction of flat interface BM after operation <50 h (Fig. 4d). For further exploration, sectional views of BMs were re-examined and apparent cracks & delamination were observed for BM with the flat interface. Oppositely, 3D interfaces with mortise–tenon joint of MBM are maintained as initial after operation over hundreds of hours. As one possible explanation, the mortise–tenon joint structure might interlock AEL and CEL layers more tightly to withstand the osmotic stress (inset of Fig. 4e), and this WD stability also offers MBM electro devices with reliability at large current and long lifespan.

The performance of MBM is further compared with BMs commercial-available or recently reported based on the transmembrane voltage drop at 100 mA cm$^{-2}$ (U100) and the 1st limiting current density (Fig. 4f and Supplementary Table 1)[18,23,30,32,35,39–47]. Contributed by mortise–tenon joint design, MBM simultaneously represents superiorities in both aspects, anticipating satisfying performances for the novel BM electro reactors that unachieved before.

## Continuous ammonia electrosynthesis

A continuous flow reactor for NH$_3$ electrosynthesis was assembled (Fig. 5a and Supplementary Figs. 27, 28) as a proof of concept, and an alkaline environment (1 M OH$^-$) for electrosynthesis was adopted for both sides to enable the non-noble metal catalysts and to surpass

hydrogen evolution reaction (HER). For demonstrating the necessity of using a BM in the established system, three kinds of ion exchange membranes were elected for comparison, in which circumstance only a BM can steadily maintain a nearly constant cell voltage with 1000 mA cm$^{-2}$ electrolytic current for 5000 s (5000 coulombs transferred), while the same mission can hardly be achieved by cation or anion exchange membranes (Fig. 5b). Meanwhile, the anodic K$^+$ and NO$_3^-$ concentration variation proved that merely BM can realize an ionic balance via WD, but oppositely severe ionic crossover occurred in cation exchange membrane (K$^+$) or anion exchange membrane (NO$_3^-$) cases, leading to instability of this process.

To achieve a high yield NH$_3$ electrosynthesis with relatively low NO$_3^-$ concentration, we specifically designed a Co 3D nanoarray self-supported catalyst by electrodeposition, where Co nanoarray was densely aligned on Co framework (Fig. 5c and Supplementary Figs. 29–32). To conquer the severe HER at high current, the principle of design is to enhance mass transfer by a multilevel structure, thus reactant NO$_3^-$ can be easily transferred through the 3D framework to Co nanoarray possessing abundant catalytic sites, which can also be proved by the depth of microporous and the bulky surface area (Supplementary Figs. 33–35, see detailed discussion in Supplementary Note 5). The unique structure of Co catalyst promised an over 90% NH$_3$ Faradaic efficiency (FE) with 2000 ppm NO$_3^-$ as well as surpassed side products (NO$_2^-$ and N$_2$H$_4$) at >1000 mA cm$^{-2}$ (Supplementary Figs. 36–43). Coupling with NiFe anode with low OER overpotential reported by us previously (Supplementary Figs. 44, 45)[48], the MBM electrosynthesis system is prospective to work with ~90% efficiency at ampere-level currents.

Through keeping identical operation conditions, I–V curves for the NH$_3$ electrosynthesis reactor using different bipolar membranes were collected as a judgment of feasibility and energy consumption. As shown in Fig. 5d, MBM reactor requested an obviously lower voltage relative to the commercial BP1 for NH$_3$ producing (especially >350 mA cm$^{-2}$), achieving 1000 mA cm$^{-2}$ under ~3 V. By deducting the thermodynamic voltage for the reaction, it could be calculated that MBM consumed lower overpotential on the generating and transporting of ions compared to BP1 (Supplementary Fig. 46). This phenomenon can be ascribed to the difference of WD ability in BMs—MBM with mortise–tenon joint interface is able to generate a necessary number of protons on time to meet the requirement for nitrate reduction, thus only consumes minor voltage increase to drive WD than BP1. Similar WD-control region can also be observed when operated with various NO$_3^-$ concentrations (from 2000 ppm to 0.1 M, Supplementary Fig 47); therefore, the emerging of MBM filled the void and anticipated a high-yield bipolar membrane NH$_3$ electrosynthesis with low electricity energy consumption at ambient conditions.

Giving credit to the fast WD of MBM and intensified mass-transfer of cathodic catalyst, the flow reactor NH$_3$ yield rate increased linearly with current density and achieved a maximum value of 68.4 mg h$^{-1}$ cm$^{-2}$ (1000 mA cm$^{-2}$) with an NH$_3$ FE of 86.2% with 2000 ppm NO$_3^-$ (Fig. 5e and Supplementary Fig. 48a), exceeding most electrochemical strategies for NH$_3$ synthesis reported so far (see comparative data in Supplementary Table S2). The same result was further proved via N-15 isotope labeling method to avoid interference from the ionic dissolution (see NMR spectrum, calibration line, and FE comparison in Supplementary Fig. 49) Main possible side product NO$_2^-$ of the system was proved insignificant in high electrolytic current (1% or less, Supplementary Fig. 52), and negligible gas product (H$_2$) can be observed produced at the cathode during operation. A higher concentration was also evaluated, and an established flow system working with 0.1 M NO$_3^-$ concentration reveals a close integrated performance compared with as low as 2000 ppm (Supplementary Fig. 48b), which is attractive for practical applications for the broader distribution from industrial effluents. Apart from these, the design of MBM reactor only consumed

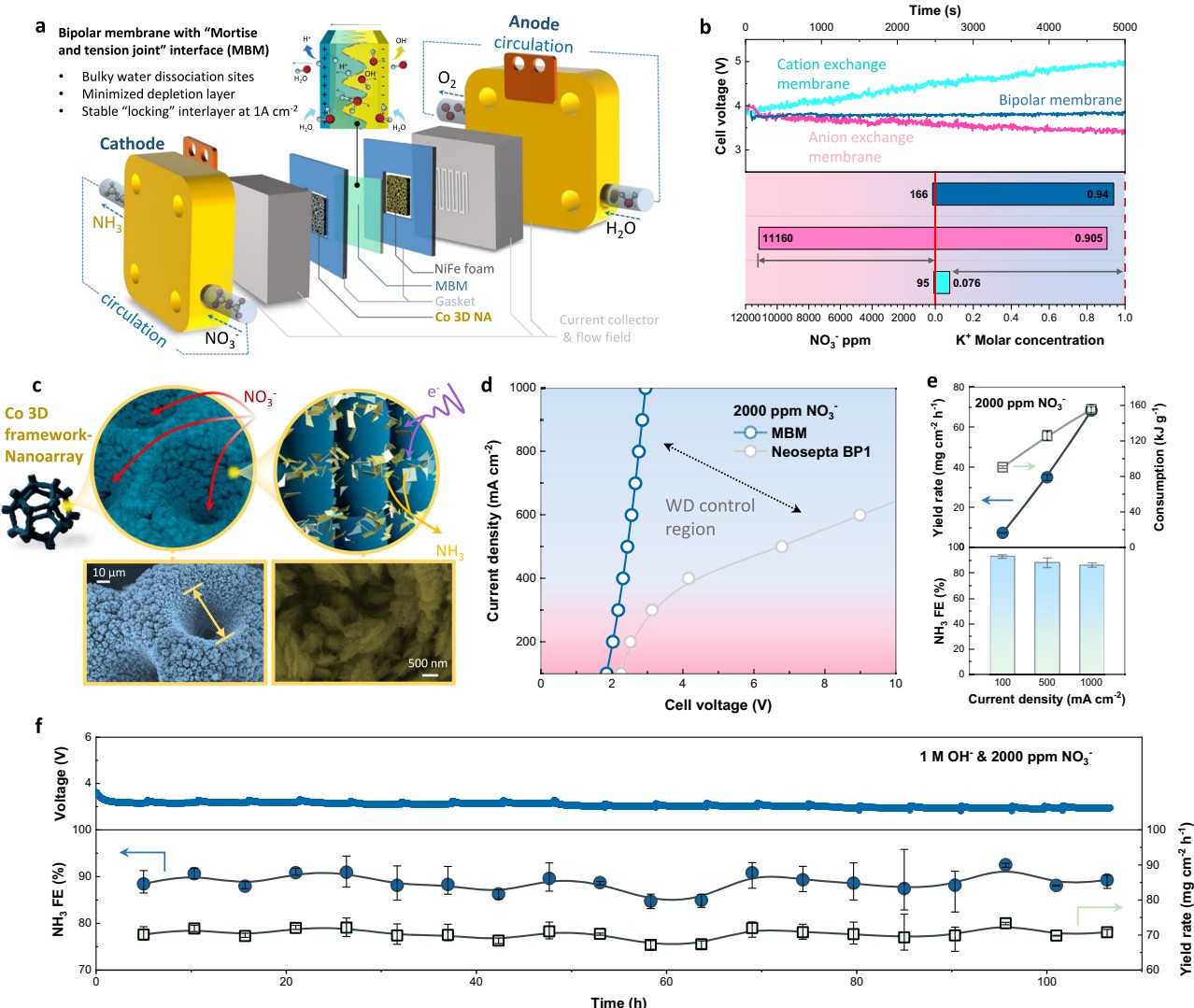

**Fig. 5 | Bipolar membrane NH₃ electrosynthesis with a flow reactor. a** Schematic of bipolar membrane reactor for NH₃ electrosynthesis in flow mode. **b** Ionic balance comparison of continuous NH₃ electrosynthesis with cation exchange membrane (Nafion 115, aqua), anion exchange membrane (Selemion DSV, pink), and bipolar membrane (MBM, mazarine) at 1 A cm⁻². The upper plot shows cell voltage variation during 5000 s electrolysis with different membranes; the lower plot presents ionic (NO₃⁻ and K⁺) concentration change of anolyte before and after electrosynthesis with different membranes. **c** Schematic illustration of Co 3D framework-nanoarray as NO₃⁻RR catalysts with intensified mass transfer for low NO₃⁻ concentration and high NH₃ yield (upper) and their SEM images of different magnifications (lower). **d** I–V curves of electroreduction NO₃⁻ to NH₃ in aqueous solutions with 1 M KOH/

2000 ppm KNO₃ in catholyte and 1 M KOH in anolyte with MBM and commercial BP1. **e** NH₃ Faradaic efficiency (lower bars), yield rates (cool aqua, left axis of upper plots) and energy consumption (pink, right Y-axis of upper plots) of MBM NH₃ electrosynthesis system (2000 ppm NO₃⁻) at a current density of 100, 500 and 1000 mA cm⁻². **f** Long-term NH₃ electrosynthesis at 1000 mA cm⁻² using MBM electrosynthesis system with 2000 ppm KNO₃/1 M KOH. The upper plots show the cell voltage versus operation time with electrolytes renewed for every 5-h electrolysis and the lower plots show NH₃ Faradaic efficiency (circular scatters, left Y-axis) and yield rates (rectangular scatters, right Y-axis) versus operating duration. Error bars denote the deviation of the yield rates calculated from three independent samples.

155.2 kJ g⁻¹ renewable electricity energy at maximum NH₃ output (1000 mA cm⁻²) at normal temperature and pressure.

Long-term stability of such a reactor operated at high current density is vital for practical applications in industrial-scale artificial nitrogen cycles. For verifying, the established reactor equipped with MBM was operated with 2000 ppm NO₃⁻ at a constant current 1000 mA cm⁻² mode for >100 h, during which imperceptible cell voltage change can be observed (Fig. 5f), in contrast into one operated with Commercial BP1 (Supplementary Fig. 51). This disparity was coincident with the WD stability as discussed before and can also be attributed by "self-interlocked" effect of the mortise-tenon joint. In the meantime, time-dependent products were also detected, revealing maintenance of ~90% NH₃ FE ~70 mg cm⁻² h⁻¹ yield rate as well as <1.5% side product NO₂⁻ over 100 h working (Supplementary Fig 53). As

predicted, long-term operation stability can also reappear in wider NO₃⁻ concentration of 0.1 M with slightly higher NH₃ yield (Supplementary Figs. 52, 53), and intact morphology of MBM interface can be maintained after long-time application at 1000 mA cm⁻² (Supplementary Fig. 54). An overall evaluation of material fabrication cost and process techno-economy (Supplementary Note. 6) also indicates 3D interfaces with mortise–tenon joint structure hopefully endows bipolar membrane with ability to withstand harsh operation conditions and delivers a promising pathway to accelerate the development of NH₃ electrosynthesis.

A universal strategy has been put forward to facilitate water dissociation (WD) kinetics and mass transfer at the bipolar membrane interface for averting the shortage of conventional BMs. Based on this principle, a BM with mortise–tenon joint interlayer (MBM) was

fabricated to enhance interfacial reaction and transportation simultaneously, realizing high WD performance in a wide current range (1.13 V transmembrane voltage at 1000 mA cm$^{-2}$) and stability (550 h). The design of MBM anticipates a continuous and high current NH$_3$ electrosynthesis from NO$_3^-$, and a Co 3D nanoarray cathode with abundant catalytic sites and intensified mass transfer was specifically fabricated to fulfill the goal. With the adoption of MBM with fast kinetics and ampere-level water dissociation ability, continuous NH$_3$ electrosynthesis can be realized over 100 h with merely 2000 ppm NO$_3^-$, achieving NH$_3$ FE of 86.2% and yield rate of 68.4 mg h$^{-1}$ cm$^{-2}$ at 1000 mA cm$^{-2}$, which exceeds most NH$_3$ synthesis process ever reported. Therefore, the NH$_3$ electrosynthesis process with the combination of novel bipolar membrane design was expected to alleviate nitrate contamination issues and partially compensate for huge global NH$_3$ consumption with normal temperature and pressure processes in the future.

## Methods

### Materials

*p*-Terphenyl (98%), 1-Methyl-4-piperidone (98%), Trifluoroacetic acid (98%), and Trifluoroacetic acid (99%) were purchased from MERYER CO. (China) and used without further purification. Dichloromethane (99.9%, Extra Dry, with molecular sieves, Water ≤ 50 ppm), Urea (99%), Dimethyl sulfoxide (99%), Iodomethane (99.5%), and dimethyl sulfoxide-d6 ((D,99.8% TMS(0.03%)) for NMR were received from Energy CO. (China). Inorganic chemicals, including sodium sulfate (Na$_2$SO$_4$, anhydrous, 99%), Cobalt chloride hexahydrate (CoCl$_2$·6H$_2$O, 99%), Nickel(II) chloride hexahydrate (NiCl$_2$·6H$_2$O, 99%), Potassium hydroxide (KOH, 85%), and Potassium carbonate (K$_2$CO$_3$, 99%) were obtained from Adamas-beta CO. (China). Perfluorinated sulfonic acid resin solution (ion exchange capacity = 0.91, 10 wt% in ethanol) for preparation of CEL (cation exchange layer) was kindly supplied by THINKRE NEW MATERIAL CO. (China). Neosepta BP1 (Tokuyama Co., Japan) was used as a high-performance representative of commercial bipolar membranes for comparison purposes. Nickel foil (thickness of 50 μm, for template growth), as well as metal foam (Cobalt and Nickel as electrodes, 1.6 mm), were bought from Alibaba Group. Deionized (DI) water (18.2 MΩ cm) was applied throughout all experiments in this research.

### Preparation of morphological CoNi array templates

All templates for membrane casting were achieved by a hydrothermal method in a typical procedure as followed: 2.38 g CoCl$_2$·6H$_2$O, 1.19 g NiCl$_2$·6H$_2$O, and 0.9 g urea were dissolved in 75 mL DI water at room temperature and filled into 100 mL hydrothermal reactor. A piece of Nickel foil was cut into 2 cm × 3 cm and placed below the surface of the metallic salt solution. The reaction proceeded in an oven at 80 °C for 3 h, and the foil substrate grown with binary metal hydroxide array was carefully taken out of the reactor after cooling down, followed by dipping into DI water for at least three times to remove the remaining precursor solution. Finally, the maroon template was dried completely at 60 °C under a vacuum for further characterization and membrane fabrication.

For preparing templates with physical parameter-regulated microstructures, metallic salt solution concentrations and hydrothermal duration for growth can be adjusted, respectively. For example, 75 mL hydrothermal precursor solutions charged with cobalt salts from 0.46 to 4.6 g (keeping precursor ratio unchanged) can realize the various thickness of the needle-like array grown on the substrate. Analogously, the height of the array can be regulated with hydrothermal duration varied from 1.5 to 4.5 h.

### Fabrication of anion exchange membrane Layer

Quaternary ammonia poly (N-methyl-piperidine-co-p-terphenyl) (QPPT) was synthesized via two steps, including Friedel−Crafts

polycondensation followed by a quaternization step[49,50]. Synthesized QPPT powder was dissolved into dimethyl sulfoxide (DMSO) to form a 5 wt% solution, and then casted onto CoNi array templates that adhere to a hot plate at 50 °C. After the fully dried solvent, the QPPT anion exchange membrane was successfully prepared interlocked with the CoNi array on one side, and kept smooth on the other side. Prepared anion exchange membrane (in iodide form), together with the substrate was soaked into 6.0 M HCl solution for 30 min, during which QPPT anion exchange membrane fell off the substrate and Metallic template was etched, followed by washing with DI water till neutral and dried in an oven at 60 °C, resulting an anion exchange membrane (in chloride form) with sub-micrometer surface structure.

### Fabrication of bipolar membranes

**Fabrication of MBM.** The prepared QPPT membrane was sticked onto a piece of the glass plate (thickness of 1 mm) with sealing tape at the edge, the remaining surface of the patterned structure upwards. SnO$_2$ powder (particle size: 30–50 nm) was dispersed uniformly into DI water ahead of time by stirring and ultrasonic treatments as catalytic ink and then loaded onto a micropattern of QPPT anion exchange membrane by spray coating method of -0.1 mg cm$^{-2}$. 10 wt% perfluorinated sulfonic acid resin solution was diluted to 1 wt% with 10:1 ethanol/DMSO (m/m) and spray coated onto an anion exchange membrane and completely covered the SnO$_2$ catalyst layer. During the coating process, the glass was heated to 80 °C to ensure that perfluorinated solution droplets were instantaneously dried when contact anion exchange membrane and avoid swelling before forming film as a cation exchange layer (CEL).

**Fabrication of flat interface BM.** The whole process of flat interface BM fabrication keeps the same to MBM, except for a prepared QPPT membrane with a smooth surface was adopted an anion exchange membrane instead of a patterned one. The thickness of membrane layers and catalyst loading for flat interface BM is to the greatest extent keep the same as the MBM.

### Basic properties measurements of AEL and CEL

Several basic properties of both AEL and CEL were measured individually, including water uptake, swelling ratio, ionic conductivities, and chemical structure.

**Water uptake (WU) measurements and swelling ratio (SR) measurements.** Measurements were applied for CEL in H$^+$ form and AEL in OH$^-$ form cut into 1 cm × 4 cm, achieving by soaking in corresponding degassed acid and alkaline solution at 60 °C for 24 h in order to change counter ions into H$^+$ or OH$^-$ form. After thoroughly washed with DI water, both samples were drastically dried and the weight ($W_{dry}$) as well as length ($L_{dry}$) of the sample in a dried state were obtained. Then the samples were wiped into DI water till fully infiltrated from 30 to 80 °C with an interval of 10 °C. Weight ($W_{wet}$) and length ($L_{wet}$) in the wet state of each temperature were recorded to calculate WU and SR of AEL and CEL by following equations:

$$WU(T)(\%) = \frac{W_{wet}(T) - W_{dry}}{W_{dry}} \times 100\% \qquad (1)$$

$$SR(T)(\%) = \frac{L_{wet}(T) - L_{dry}}{L_{dry}} \times 100\% \qquad (2)$$

**Ionic conductivity measurements.** In-plane conductivity measurements were conducted for CEL in H$^+$ and for AEL in OH$^-$ form, which were prepared as described before. Both AEL and CEL samples were cut into 1 cm × 4 cm and equipped into a four-electrode (Pt) cell for a.c.

impedance tests. The cell was placed into a degassed DI water the immersed the membrane samples and the impedance measurements were carried out at galvanostatic mode from 100 kHz to 100 Hz on PARSTAT 3000A potentiostat (AMETEK, USA) to obtain in-plane Ohmic resistance ($R$). Ionic conductivity ($\sigma$) can be obtained via the following equation:

$$\sigma = \frac{L}{R \times T \times W} \tag{3}$$

where $L$, $T$, and $W$ represent for length, thickness, and width of the measured samples.

**Chemical structure analysis.** Chemical structure of QPPT has been determined through $^1$H Nuclear Magnetic Resonance ($^1$H NMR) by Bruker AVANCE III (Germany), the sample was dissolving in dmso-d6 with ~5% TFSA added to drive $H_2O$ singlet to high chemical shift. Fourier transform infrared spectroscopy (FTIR) has also been conducted to detect the main functional group of QPPT, using Nicolet iS50 (Thermo Scientific, USA) at room temperature. Potassium bromide (KBr) was applied as the background medium and the pellet technique was adopted for the tested sample. The spectrum was obtained in the 600–4000 cm$^{-1}$ wavenumber range.

## Microscopy analysis

**Field-emission scanning electron microscope (FE-SEM) analysis.** The FE-SEM analysis was conducted for samples including the surface and cross-section of CoNi templates grown on a substrate, the surface and cross-section of QPPT anion exchange membrane with microstructure on the surface as well as bipolar membranes using Merlin Compact (ZEISS, Germany). Among cross-section samples, CoNi templates were prepared by a scratch of the lancet, and the membrane was snapped in liquid nitrogen. Energy dispersive spectroscopy detector accessory (EDS, X-max 80, OXFORD) was applied to analyze the element distribution of the cross-section of membrane samples. Low magnification SEM images were used for analyzing the pore depth, areal porosity (ImageJ software), and size distribution of the CoNi array (python based on gray level of image).

**Three-dimensional white light interference surface topography (3D-WLI, ZygoLamda NeXView, USA) analysis.** The 3D height and weight information of the microstructure for both CoNi templates and the microstructure of the anion exchange membrane were analyzed through 3D-WLI at room temperature under atmospheric pressure. The 3-D coordinates of the surface structure were used for the analysis of the CoNi array length distribution (python, see coding in Supplementary Note 3).

**Computed micro-X-ray tomography (micro-CT, Zeiss Xradia Versa 520, Germany) analysis.** Projection images of two dimension and three-dimensional interlayer structures of both MBM(s) and flat interface BMs were collected by micro-CT and reconstructed with a resolution of 0.5 μm per voxel.

## Electrochemical impedance spectroscopy (EIS) analysis of bipolar membranes

The EIS experiments were conducted with a self-made four-electrode cell equipped with MBM, flat interface BM as well as Neosepta BP1. As shown in Supplementary Fig. 20, bipolar membranes were stuck in the middle of two symmetrical compartments with an effective area of 1 cm$^2$. Two Platinum electrodes placed outboard act as working electrodes and counter electrodes. Two reference electrodes were respectively placed inside the Luggin capillary that contacts the surface of bipolar membranes. The cell was charged with 0.5 M Na$_2$SO$_4$ solution and should stand for 24 h to reach a steady state before EIS

data collection. The data was obtained by PARSTAT 3000A potentiostat (AMETEK, USA) in galvanostatic mode with a frequency from 100 kHz to 0.1 Hz. An appropriate amplitude should be set to minimize the signal noise and keep data accuracy as well. The acquired Nyquist plot was fit by Z-view software based on the equivalent circuit (Inset of Fig. 3g). Five parameters to describe ionic transportation and reaction kinetics at the interface of the bipolar membrane, including ohmic resistance, ionic transportation, water dissociation constant, the capacitance of depletion layer can be derived based on protonation-deprotonation mechanism[30].

## Adhesion strength measurements

Stripping strain has been measured to evaluate the adhesion strength of AEL and CEL using Universal Material Testing System (HZ-1004B, Hengzhun, China) at room temperature. Samples of the bipolar membrane in the wet state with the size of 1 cm × 4 cm were prepared and the strong sticky tape was put onto each side of the membrane layers (AEL and CEL), followed by being clamped into the upper and lower part of the testing fixture and tested with a strain rate of 1 mm/min.

## Measurements of bipolar membrane water dissociation performance and stability and selectivity

**Determination of bipolar membrane water dissociation performance at low current density (0–100 mA cm$^{-2}$).** Bipolar membranes were equipped in a four-electrode cell with Luggin capillary design (Supplementary Fig. 20). Membranes with a 1 cm$^2$ effective area were left to stand in a cell for 24 h before data was recorded. Two platinum plate electrodes were connected to the working and counter electrodes and two Ag/AgCl electrodes in the Luggin capillary were connected to sensor and reference electrodes. Current–voltage ($I$–$V$) curves were recorded as a performance indicator, which was obtained using PARSTAT 3000A potentiostat (AMETEK, USA) under galvanodynamic setup with a current range from 0 to 100 mA and a rate of current change of 2 mA/s and $I$–$V$ relationships were collected. All tests were conducted in 0.5 M Na$_2$SO$_4$ at room temperature, and experiments were cut off when scanning to 100 mA or reaching the upper limit of potentialstat applied voltage.

**Determination of bipolar membrane water dissociation ability at high current density (0–1000 mA cm$^{-2}$).** Same four-electrode cell as described were adopted to collect $I$–$V$ data at a high current density, and pre-treatment are kept the same as adopted in low current density experiments with 0.5 M Na$_2$SO$_4$ or 1 M KOH or 1.0 M acid/base unsymmetrical electrolytes. Here a programmable current plant SS-L605SPD (A-BF, China) was connected to the working and counter electrode (Platinum) of the cell to avoid the applied voltage limitation, and two reference Ag/AgCl electrodes were connected to the potentialstat. The current plant was programmed to keep current across the bipolar membrane increasing by 2 mA/s from 0 to 1000 mA, and the transmembrane voltage drop was recorded by potentialstat at open current voltage mode. $I$–$V$ curve of large current range can be obtained by simple transformation of recorded voltage versus time and linear relationship between current and time. During experiments, the electrolytes in both compartments were continuously refreshed with peristaltic pumps at a flow rate of 72 mL min$^{-1}$ to minimize the concentration polarization effects.

**Water dissociation stability assessments.** Water dissociation stability of both MBM and flat interface BM was evaluated under a galvanostatic mode with same four-electrode cell as described before. A current of 300 mA was kept in the cell by a current plant DP 3020 (MESTEK, China) through two platinum electrodes and transmembrane voltage drop versus time was recorded by a battery testing system CT3002A (LANHE, China) under pending mode. During stability tests,

electrolytes on both sides were fully circulated and refreshed to minimize the effects of polarization.

**Determination of bipolar membrane selectivity.** Selectivity of MBM, flat interface BM and commercial Neosepta BP1 was indicated by 1st limiting current density of water dissociation, which can be derived through the first-order derivative of $I-V$ relationship versus current, which the value of $X$ when the dependent variable (d$V$/d$I$) reaches its maximum value.

**Performance of bipolar membrane NH$_3$ electrosynthesis via NO$_3^-$**

**Fabrication of Co 3D nanoarray catalyst for NO$_3^-$RR.** The 3D Nanoarray of Co was fabricated as a self-supported catalyst via electrodeposition strategy. The commercial Co foam was selected as the substrate, which is pre-cleaned by ultra-sonification in 6 M HCl for 10 min. The 1st step is to construct a Co 3D framework via a rapid H$_2$ template-electrodeposition method on the substrate, which is conducted using a two-electrode setup in electrolytes containing 0.1 M CoCl$_2$ and 1 M NH$_4$Cl at 2 A cm$^{-2}$ for 600 s. After rinsing in DI water several times, the 2nd step is to construct Co nanoarray on the framework. A three-electrode setup was used with an electrolyte containing 0.05 M Co(NO$_3$)$_2$. The electrodeposition was firstly at −1.0 V vs. saturated calomel electrode (SCE) for 1200 s and then reduced in 1 M KOH at −1.4 V vs. Hg/HgO for 600 s before washing with DI water.

For comparison, Commercial Co foam, Co nanoarray, and Co 3D framework were also tested for their NO$_3^-$ performance, respectively. The Co foam was used as a commercial material without any further treatment, and the Co nanoarray was obtained by electrodeposition on Commercial Co substrate (same as the second step of constructing Co 3D nanoarray). Co 3D framework was obtained as the 1st step of constructing the Co 3D nanoarray catalyst.

**Catalytic performance measurements of electrodes.** The $I-V$ curves of Co-based catalysts for NO$_3^-$RR and hydrogen reduction reaction (HER) were conducted using PARSTAT 3000A potentiostat (AMETEK, USA) in 1 M KOH alkaline electrolytes and the ones containing 2000 ppm or 0.1 M KOH, respectively. A 1 cm × 1 cm of self-supported catalysts was equipped as the working electrode of three-electrode measurement, and the Hg/HgO was selected as the reference electrode. The EIS measurements were conducted from 100 kHz to 1 mHz at −1 V vs. reference electrode. Cyclovoltammetry (CV) measurements for double-layer capacitance and ECSA calculation were conducted in the same conditions from 20 to 100 mV s$^{-1}$ for each catalyst.

The faradaic efficiency and yield rate of main product NH$_3$ and side products NO$_2^-$ was determined by the colorimetric method as discussed below. The samples were taken from the electrolytes after constant potential electrolysis for 30 min.

NiFe on Nickel foam was selected and fabricated as an anode electrode for oxygen evolution reaction (OER)[48]. The catalytic performance was evaluated by three-electrode methods, including a 1 cm$^2$ NiFe working electrode, a graphite counter electrode and a Hg/HgO reference electrode. The testing was conducted in 1.0 M KOH electrolyte by linear sweep voltammetry method using PARSTAT 3000 A potentiostat (AMETEK, USA) at a scan rate of 5 mV/s. All the potentials were converted into potential vs. RHE according to Nernst equation. EIS experiment for catalytic electrode was conducted under same condition by potentialstatic EIS mode with the frequency from 100 kHz to 1 mHz.

**Determination of NH$_3$ Faradaic efficiency, yield rate and energy consumption at different current density.** The flow cell was working at galvanostatic mode using current plant DP 3020 for 30 min and the

electrolyte samples at each current were collected for NH$_3$ detection. The total volume of both cathode and anode electrolyte circulated is 250 mL for experiments at every current density. The electrolytes at each side were refreshed between every individual test. NH$_3$ Faradic efficiency, yield rate, and energy consumption can be calculated based on the following equations:

$$NH_3 \text{ Faradic efficiency} = \frac{\left(8 \times F \times C_{NH_3} \times V\right)}{I \times t} \quad (4)$$

$$NH_3 \text{ Yield rate} = \frac{C_{NH_3} \times V \times 17}{t \times S} \quad (5)$$

$$\text{Energy consumption} = \frac{I \times U \times t}{C_{NH_3} \times V \times 17} \quad (6)$$

where $F$ is the Faraday constant (96,485 C mol$^{-1}$), $C_{NH_3}$ is the measured NH$_3$ concentration, $V$ is the volume of the cathodic electrolyte, $I$ is the electrolysis current applied, $t$ is the electrolysis duration, $S$ is the effective surface area of electrodes and $U$ is the corresponding applied voltage to the flow cell, which can be found in polarization curve.

**Determination of Faradaic efficiency, yield rate at different current density of possible side products in catholyte.** As discussed, NO$_2^-$ and N$_2$H$_4$ are the main products accompanied with NO$_3^-$RR. The samples to be measured are taken the same way as mentioned above. Faradic efficiency, yield rate and energy consumption of NO$_2^-$ and N$_2$N$_4$ can be calculated based on the following equations:

$$NO_2^- \text{ Faradic efficiency} = \frac{\left(F \times C_{NO_2^-} \times V\right)}{I \times t} \quad (7)$$

$$KNO_2 \text{ Yield rate} = \frac{C_{NO_2^-} \times V \times 85}{t \times S} \quad (8)$$

$$N_2H_4 \text{ Faradic efficiency} = \frac{\left(14 \times F \times C_{N_2H_4} \times V\right)}{I \times t} \quad (9)$$

$$N_2H_4 \text{ Yield rate} = \frac{C_{N_2H_4} \times V \times 17}{t \times S} \quad (10)$$

where $F$ is the Faraday constant (96,485 C mol$^{-1}$), $C_{NO_2^-}$ and $C_{N_2H_4}$ are the concentration of NO$_2^-$ and N$_2$H$_4$, $V$ is the volume of the cathodic electrolyte, $I$ is the electrolysis current applied, $t$ is the electrolysis duration, $S$ is the effective surface area of electrodes.

**Continuous bipolar membrane NH$_3$ electrosynthesis.** A flow cell for NH$_3$ electrosynthesis was composed of endplates, current collectors, flow fields, gaskets, electrodes, and bipolar membranes, with electrolytes circulation powered by a double channel peristaltic pump as shown in Fig. 5a and Supplementary Figs. 28, 29. Co 3D nanoarray (Supplementary Figs. 29–35) and the as-prepared NiFe foam[48] were adopted to be cathodic and anodic catalysts. The effective working area for both anode and cathode electrodes is 3.0 cm$^2$, respectively, sandwiching a bipolar membrane slightly larger to avoid electrolytes convection on both sides. The cell was assembled in sequence and then fastening by screws with a torque wrench at 5 N m. 2000 ppm (or 0.1 M, Supplementary information) KNO$_3$/1 M KOH and 1.0 M KOH solutions were flowed into the

cathode and anode, respectively, at a speed of 72 mL/min for 1 h before testing.

A certain current was applied by current plant DP 3020 and kept for 1 min to reach steady state, and overall voltage applied to the flow cell was collected. $I–V$ curves for MBM were acquired at room temperature in the current density range 0–1000 mA cm$^{-2}$ (0–650 mA cm$^{-2}$ for BP1 for the upper limiting of device). The electrolytes were refreshed between two independent testing under different currents to avoid product influences. Further determination of Faradaic efficiency, yield rate and systematic stability was described in the Supplementary information.

**NH$_3$ electrolysis stability analysis.** The stability of bipolar membrane NH$_3$ synthesis flow cell system was tested through an intermittent mode with both cathode and anode electrolytes renewed for every 5 h. Electrolytes circulated in each step were kept to 100 mL with a circulating speed of 72 mL/min, and a small portion of cathodic electrolytes are taken as samples to be measured for figuring out Faradic efficiency, yield rate, and energy consumption change of NH$_3$ and other side products versus operation time.

**Production detection.** The concentration of produced NH$_3$ was determined by spectrophotometrically method using indophenol blue method. The cathodic samples should be diluted before detection for their high concentrations. In a typical way, 2 ml of diluted samples were added with 2 ml of a 1 M NaOH solution that contained salicylic acid and sodium citrate, followed by instilled 1 ml of 0.05 M NaClO and 0.2 ml of 1 wt% $C_5FeN_6Na_2O$ (sodium nitroferric cyanide). After being placed for 2 h, the color of the mixed sample was detected under ultraviolet–visible spectrophotometer and from 500 to 800 nm and the concentration of NH$_3$ was determined using the absorbance at a wavelength of 652 nm according to the calibrated line obtained beforehand.

A typical Griess test was adopted for NO$_2^-$ concentration determination[51]. To prepare Griess reagent, 4 g of N-(1-naphthyl) ethyldiamine dihydrochloride, 0.2 g of ethylenediamine dihydrochloride and 5 mL of H$_3$PO$_4$ were dissolved in 25 ml of deionized water. 5 ml of catholyte samples were diluted to proper concentration and adjusted to neutral pH for colorimetric detection and 0.1 mL of prepared Griess reagent was added. After 30 min of shaking, 400–700 nm absorbance of samples was measured by UV spectroscopy and 540 nm was used for concentration determination according to the calibration line.

The concentration of N$_2$H$_4$ was detected by Watt and Chrisp's method as reported[52]. P-(dimethylamino) benzaldehyde (4 g), HCl (concentrated, 30 ml), and ethanol (300 ml) were mixed to prepare chromogenic reagent and 0.1 mL was added into 5 mL diluted catholyte samples (adjusted to pH = 3 with H$_3$PO$_4$) to be detected. 400–550 nm absorbance was collected and N$_2$H$_4$ concentration was calculated with absorbance at 460 nm according to the calibration line.

**Isotope labeling experiments.** We selectively conducted isotope labeling experiments under NMR measurements to verify the reliability of obtained NH$_3$ Faradaic efficiency. Firstly, we conducted NH$_3$ electrosynthesis with the same materials and conditions but replace the $^{14}$N-KNO$_3$ by 99% (atom) $^{15}$N-NO$_3$. 500 µL of electrolytes was taken out and neutralized to weak acid by 2 M HCl as a sample and mixed with D$_2$O to achieve a total amount of 600 µL. The mixed electrolyte sample of $^{14}$N and $^{15}$N was qualitatively detected by $_1$H NMR (Bruker, 400 MHz), and different peak splitting of H can be distinguished. Secondly, a $^{15}$N-NH$_4$Cl calibration line was obtained for further determination of NH$_3$ Faradaic efficiency. We prepared several concentrations of $^{15}$N-NH$_4$Cl solutions (~90% H$_2$O and 10% D$_2$O, adding 2 M HCl till weak acid) with the precise amount of

maleic acid as an external standard. The samples were detected under NMR and the concentration of $^{15}$N could be indicated by the peak area ration between $^{15}$NH$_4^+$ and external standard so that a calibration line can be obtained. Thirdly, the NH$_3$ electrosynthesis at different current densities (200, 400, 600, 800, 1000 mA cm$^{-2}$) was conducted with 2000 ppm $^{15}$N-KNO$_3$, and the NMR samples were prepared with the same method. The concentration of NH$_3$ could be quantitatively determined according to the calibration line, and the Faradaic efficiency could be calculated and compared with the UV–Vis method.

## Data availability
The data supporting the findings of this study are available within the paper, Supplementary Information, and Source Data files. Further data beyond the immediate results presented here are available from the corresponding authors upon reasonable request.

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

## Acknowledgements

This work was supported by the National Natural Science Foundation of China (22278239) and the National Key R&D Program of China (2020YFB1505602 and 2018YFE0202001). The authors thank Z. Zhang and C. Jiang for their help in the COMSOL simulation and discussion.

## Author contributions

B.G.W. conceived and designed the research project. Z.A.X., L.W., and Y.W.L. synthesized all chemical compounds, collected and analyzed the data. Z.A.X., M.B.P., and Y.W.L. conducted electrochemical measurements and other basic property measurements. Z.A.X., Q.X., L.W., and P.C.W. assembled the bipolar membrane NH₃ flow cell and completed data collection. B.G.W., Z.A.X. wrote the paper. All the authors discussed the data and commented on the paper.

## Competing interests

The authors declare no competing interests.
