## [Peer review file · Nature Communications]

Reviewers' comments:

Reviewer #1 (Remarks to the Author):

In this manuscript, the authors showed a design of bipolar membrane with mortise-tenon joint interlayer for nitrate electroreduction. Benefiting from the promotion of mass transfer and the inhibition of HER, this strategy achieved good ammonia yield rate and catalytic stability. In addition, it has certain universality. It can be potentially considered for publication after addressing the following comments.

1. For the fabrication of AEM, the CoNi hydroxide array together with the substrate was soaked into 6.0 M HCl solution for 30 min. The solution should also etch CoNi hydroxide. How can CoNi hydroxide remain on the AEM after this process?
2. What are the reasons for the continuous decline of ammonia production rate and Faraday efficiency and the increase of energy consumption in Figure 5?
3. The performance of bipolar membrane was tested in neutral condition (Na₂SO₄ solution), but the electrocatalytic nitrate reduction was carried out in alkaline solution. Please give an explanation or provide the important tests for the performance of the membrane under alkaline conditions.
4. The authors do not seem to be concerned with the design of the catalyst. Therefore, we cannot distinguish whether the current attenuation is affected by catalyst or membrane. Therefore, some necessary control experiments should be carried out regarding catalyst performance and stability.
5. Does the bipolar membrane peel off after long-term stability test?
6. The stability provided by the authors is very good. But the electrode area of 1 cm² is too small for industrialization. The authors should do the experiment with larger electrode area.
7. Please explain in detail the specific principle and mechanism of double electrode membrane used in nitrate reduction, which should include each step of reaction.
8. The authors statement "a bipolar membrane flow reactor for continuous NH₃ electrosynthesis is realized at 1000 mA cm⁻² with Faradaic efficiency of 94.7%, state-of-art yield rate of 75.1 mg h⁻¹ cm⁻² and stable operation durability of 100 hours." However, in the figure 5, the FE may be low than 90% after 30 h. Therefore, the author's statement is not rigorous enough and should be revised.

9. It is not uncommon to realize large current at 1 M nitrate solution. Thus, some reports of high current density should also be summarized and supplemented.

Reviewer #2 (Remarks to the Author):

The manuscript explores the use of a bipolar membrane to synthesize ammonia from nitrate. The work relies on unrealistic conditions of treatment of high alkaline solutions containing high nitrate concentrations, which would fail to provide the solution of ammonia recovery outlined in the introduction. The manuscript contains many high-sounding but empty words, which results in a tedious lecture with very unclear message. The manuscript seems to be mostly focused on the study of membranes while using commercial electrodes that do not seem to be at the state of the art. Authors fail to identify key researchers on this hot-topic of N-economy and electrosynthesis of ammonia from nitrate. Indeed, language and description used falls far from the recommended in the field of ammonia electrosynthesis from nitrate. At some point it is quite unclear what is the research question or advance being proposed and mixes concepts of completely different electrified processes. The manuscript may be a better fit for a specialized engineering journals with focus on reactor design/behavior that report systematic studies. Schematics are unclear and do not depict the details or key aspects, some of them are even far from the experimental observations and extrapolate a subjective idea (do not describe observed characteristics of synthesized materials/membranes). Unfortunately, acceptance of the manuscript cannot be encouraged.

Reviewer #3 (Remarks to the Author):

The manuscript reported the design and use of bipolar ion exchange membrane for nitrate electroreduction. They claim that the highest current density can reach 1000 mA cm⁻², and the continuous ammonia production would exceed 100 hours. This is attractive data, but not novel. Similar achievements have been reported by other groups (See a recently published article: Adv. Sci., 2021, 8, 2004523. Metallic Co Nanoarray Catalyzes Selective NH₃ Production from Electrochemical Nitrate Reduction at Current Densities Exceeding 2 A cm⁻²). In addition, the exploration of the mechanism and the design of the catalyst are mediocre.

Therefore, this manuscript is not suitable for being published on Nature Communications.

1. Are Supplementary Fig 9e and 10e the same? There are many similar situations. Although the author may try to facilitate comparison, it will be very confusing to the readers.
2. Error bars should be added for important experimental data. In addition, the results are repeated for three times. Otherwise, the conclusion may be inaccurate.
3. Why did the author choose NiCo LDH as the model catalyst? LSV measurements showed that the maximum current density was only 0.35 A cm^{-2} at -0.65 V vs. RHE . This means that the actual applied voltage at -1 A cm^{-2} is more negative. A large number of research results show that the catalyst will undergo in-situ conversion at a very negative potential. Therefore, the stability of the catalyst after electrochemical test needs to be reconsidered. In addition, the long-term test results showed that the performance of the catalyst had declined. Is this caused by catalyst conversion or membrane changes?
4. If the catalyst is converted in situ, a small amount of dissolved Ni or CO may interfere with the UV-vis results. Therefore, other quantitative methods of ammonia should be provided. NMR tests are recommended.
5. The highlight of this paper is the selection of bipolar membranes to promote the migration of anions and cations. Therefore, the pH of the solution in the anode and cathode cells after electrochemical testing is an important parameter, especially after high current density and long-term electrochemical testing. The authors should mention these details.
6. The authors are encouraged to provide the original data.
7. To facilitate researchers to compare these results, ECSA normalized data should be provided to exhibit the intrinsic activity of this catalyst. The author summarize the recent reports so that the author and readers can understand the progress of nitrate electroreduction.
8. Some grammatical errors should be double-checked.

Response to reviewers

Manuscript NCOMMS-22-31417-T

We quietly appreciate the editor for offering us an opportunity to resubmit our manuscript. We also greatly thank the editor and all reviewers for your time spent making your constructive remarks and useful comments, which has significantly raised the quality of the manuscript and has enable us to improve the manuscript. Each suggested revision and comment were accurately incorporated and considered. Below the comments of the reviewers are responded point by point and the revisions are indicated.

Reviewer 1: page 2-16

Reviewer 2: page 17-21

Reviewer 3: page 22-36

Reviewer 1:

General Comments:

In this manuscript, the authors showed a design of bipolar membrane with mortise-tenon joint interlayer for nitrate electroreduction. Benefiting from the promotion of mass transfer and the inhibition of HER, this strategy achieved good ammonia yield rate and catalytic stability. In addition, it has certain universality. It can be potentially considered for publication after addressing the following comments.

Reply:

The high and positive comments are appreciated.

Comment 1:

For the fabrication of AEM, the CoNi hydroxide array together with the substrate was soaked into 6.0 M HCl solution for 30 min. The solution should also etch CoNi hydroxide. How can CoNi hydroxide remain on the AEM after this process?

Reply 1:

Thank you very much for sharing your concerns about the remaining of CoNi hydroxide. Here I would like to response for your worriers in a more detailed way. Actually, **the only function of CoNi hydroxide nanoarray is to act as a template**, so that surface topology of AEM can be formed after sacrificing in 6.0 M HCl. As your correctly understanding, CoNi was etched together with the substrate, and the grass-like AEM surface (also polymer materials) can be exposed during fabrication process, and subsequent contents of this work did not involve CoNi materials anymore. For example, the interlayer WD catalyst adopted is SnO₂, and the electrocatalytic materials adopted for NO₃⁻RR and OER are metal Co and NiFe self-supported electrodes, respectively.

For express more precisely, we revised the expression in the manuscript (page 5) and copied here:

As shown in Figure 2b, the template was firstly prepared by self-growth of CoNi hydroxide on metal substrate under hydrothermal method to obtain a needle-like microarray, and then QPPT ionomer solutions was casted on this template surface for AEL fabrication (see chemical structure in Supplementary Figs. 7-8). After peeling off from the substrate and etching the embedded microarray, the AEL with micro-patterned surface was obtained. Finally, the fabrication of MBM was completed by spray-coating of WD catalysts ink and PFSA CEL on patterned side of AEL, hereto “mortise-tenon joint” structure was achieved.

We would also add new annotations in the Figures (Figure R1) as followed for your better understanding:

Figure R1. a, CoNi template in nanoarray form growing on substrate; **b**, Exposed AEM surface with grass-like topology after etching templates.

Comment 2:

What are the reasons for the continuous decline of ammonia production rate and Faraday efficiency and the increase of energy consumption in Figure 5?

Reply 2:

Thank you for the rigorous query about the performance decline of the established NH_3 electrosynthesis reactor in the last version of manuscript. After chewing over this issue, we reasonably speculated that **the NH_3 Faradaic efficiency and yield rate decline can be caused by the product crossover** (In this work we took the NH_3 concentrations of cathode side into account for systematic FE calculations). The NH_3 diffused cross the bipolar membrane driving by concentration difference can lead to NH_3 reduction in catholyte, causing decrease in the calculated FE and yield, as well as increase in average energy consumption for per gram of NH_3 . Another secondary evidence is that the cell voltage becomes lower after working for tens of hours, which can also be resulted by a part of NH_3 crossover from cathode and re-oxidized at the anode. Therefore, we try to attribute the root of the problem to the membrane layers, which enjoys low ionic resistance at the expense of dimensional stability in aqueous system.

For solving this problem, **we added a PTFE substrate for the anion exchange layer to reinforce the membrane and to control the swelling**. The substrate we adopted is not only chemically stable, but also possess porosity over 90%, which result negligible influence for the original performance of MBM. The updated stability data showed that NH_3 faradaic efficiency and yield rate can be well maintained for over 100 hours (no matter with 0.1 M or 2000 ppm NO_3^-), as shown in Figure R2.

Figure R2. Stability measurements of MBM (reinforced) NH_3 electrosynthesis with a, 2000 ppm and b, 0.1 M NO_3^- . The upper plots show the cell voltage variation and the lower one showed the NH_3 Faradaic efficiency (circular scatter-line, left Y-axis) and corresponding yield rates (rectangular scatter-line, right Y-axis).

And we also revised the involved expression in the manuscript as followed (page 5):

As predicted, long-term operation stability can also be realized in wider NO_3^- concentration of 0.1 M with slightly higher NH_3 yield (Supplementary Figs. 50-51), and intact morphology of MBM interface can be maintained after long time application at 1000 mA cm^{-2} (Supplementary Fig. 52)

Moreover, the maintenance of new nitrate reduction catalyst and OER catalyst after 100 hours working were shown in the Reply 4, which might further answer your concern.

Comment 3:

The performance of bipolar membrane was tested in neutral condition (Na_2SO_4 solution), but the electrocatalytic nitrate reduction was carried out in alkaline solution. Please give an explanation or provide the important tests for the performance of the membrane under alkaline conditions.

Reply 3:

Thank you for your constructive suggestions on the pH conditions we adopted, which is helpful for us to perfect the logic of the manuscript. Following your advice, we supplement relative experimental evidence in the Supplementary information (Supplementary Fig. 26), **which**

presents the performance in 1M KOH symmetrical system (Figure R3) to verify the water dissociation ability in alkaline system.

Figure R3. I-V curves of MBM in current range of **a**, 0-200 mA cm⁻² and **b**, 500-1000 mA cm⁻².

In light of measurement, we amazingly find that a higher difference between the performance of MBM and commercial Neosepta BP1 was revealed, especially at high current range. MBM maintain a WD performance close to in the near pH (0.5 M Na₂SO₄), consuming ~1 V across the membrane at ~1000 mA cm⁻². However, the WD kinetics of commercial BP1 further retarded in alkaline pH. We find the result can reasonably explain why the flow reactor for alkaline NH₃ synthesis revealed higher performance discrepancy than for neutral WD experiments when equipped with the two kinds of bipolar membranes. We speculate that the extra performance decline of BP1 can be attributed to the surface pH value change of interlayer catalyst (*Science*369, 1099–1103 (2020)), while WD promoted by “mortise-tenon joint” can avoid this flaw.

The relatively expression in the manuscript was revised (page 11) as followed:

As a potential candidate for the emerging electrochemical devices with electrolytes far from neutral (e.g. alkaline NH₃ electrosynthesis discussed below) or even unsymmetrical (e.g., 1 M H⁺/OH⁻), MBM also reveals superiority of energy efficiency for WD compared to commercial BP1 as shown in Figure 4c and Supplementary Fig. 26.

Comment 4:

The authors do not seem to be concerned with the design of the catalyst. Therefore, we cannot distinguish whether the current attenuation is affected by catalyst or membrane. Therefore, some necessary control experiments should be carried out regarding catalyst performance and stability.

Reply 4:

We unfeignedly think the catalyst-related advice is quite beneficial for us to promote this work. As far as we can understand, the “catalyst” mentioned here can possibly be referred to i) water dissociation catalyst at bipolar membrane interlayer or ii) the cathodic catalyst for nitrate reduction reaction that we adopted in the flow cell. (NiFe catalyst for OER was reported in our previous work with detailed experimental data (*ACS Applied Energy Materials* 4, 9022-9031 (2021)).)

For explanation of situation i): The WD catalyst we adopted in this manuscript for fabricating both “mortise-tenon” bipolar membrane or “flat” bipolar membrane is SnO₂ nanoparticle, a kind of commonly used and high-performance WD catalyst. Hence, the only difference of both membranes is the interfacial morphology, and we believe the performance discrepancy between two bipolar membranes could be reasonably attributed to the interlayer structure.

For the situation ii): To resolve the concerns from the reviewer and also to realize a better NH₃ electrosynthesis performance with lower concentration (2000 ppm or 0.1 M), **we designed and fabricated a series of new catalysts with different morphologies with merely Co element.** Co foam that we adopted in the last version of manuscript were also included as a control and comparison. Based on the individual performance for NO₃⁻RR process in same condition, we selected the best one to act as the final cathodic catalyst and re-conducted all the subsequent NH₃ flow cell experiments.

To be specific, we fabricate three NO₃⁻RR self-supported catalysts other than Co foam (named Co nanoarray, Co 3D framework and Co 3D nanoarray), and the LSV as well as EIS measurements are shown as below (Figure R4):

Figure R4. Linear sweep voltammetry curves and EIS measurements of **a,b**, Co foam; **c,d**, Co Nanoarray; **e,f**, Co 3D framework; **g,h**, Co 3D Nanoarray. The measurements were conducted with 1 M KOH with none (HER), 2000 ppm KNO_3 or 0.1 M KNO_3 (for LSV only), respectively.

We find from the supplementary data that all catalysts constructed with Co element showed a far higher activity compared to hydrogen evolution reaction from the LSV and EIS measurements. In the meantime, Co 3D nanoarray can realized a highest NO_3^- RR current at the same potential applied compared to the other ones and can still achieve $> 1000 \text{ mAcm}^{-2}$ with NO_3^- concentration as low as 2000 ppm (please find more detailed information and discussion in new version of Supplementary figures and Note 5). Consequently, we selected the Co 3D nanoarray from the candidate as the cathodic catalyst and all the experiments in the “ NH_3

electrosynthesis part” were retested by equipping new catalyst into the established bipolar membrane flow system. All the renewed data (including performance and stability) was completed with this best catalyst, so we may reasonably put down the difference between experimental group and control group of the current stage to the bipolar membranes.

Here shares the relevant revision as followed (page 13):

To achieve a high yield NH_3 electrosynthesis with relatively low NO_3^- concentration, we specifically designed a Co 3D nanoarray self-supported catalyst by electrodeposition, where Co nanoarray was densely aligned on Co framework (Figure 5c and Supplementary Figs. 29-32). To conquer the severe HER at high current, the principle of design is to enhance mass transfer by a multilevel structure, thus reactant NO_3^- can be easily transferred through 3D framework to Co nanoarray with high intrinsic reduction activity, which can also be proved by the depth of microporous and the bulky surface area (Supplementary Figs. 33-35, see detailed discussion in Supplementary Note 5). The unique structure of Co catalyst promised an over 90% NH_3 Faradaic efficiency (FE) with 2000 ppm NO_3^- as well as surpassed side products (NO_2^- and N_2H_4) at $>1000 \text{ mA cm}^{-2}$ (Supplementary Figs 36-42). Coupling with NiFe anode with low OER overpotential reported by us previously (Supplementary Figs. 43-44)⁴⁸, the MBM electrosynthesis system is prospective to work with high efficiency at ampere-level.

To further put away your worriers for the stability of catalysts adopted in this work, we also compare the performance maintenance of the adopted ones after used for >100 hours. **As shown in the Figure R5, the decline is acceptable for both catalysts.**

Figure R5. Performance decline after over 100-hour NH_3 electrosynthesis. **a**, cathodic catalyst; **b**, anodic catalyst.

Comment 5:

Does the bipolar membrane peel off after long-term stability test?

Reply 5:

Thank you for raising this problem that strongly associated with the core novelty of this work. **The bipolar membrane does not peel off after the long-term stability test.** It can be confirmed by the below Figure R6 showing that the cross-section morphology of one sample after over 100 hours of NH_3 electrosynthesis. We also supplement relevant information in SI (Supplementary Fig. 52):

Figure R6. SEM image of MBM interface morphology after NH_3 electrosynthesis.

And we made the corresponding revision (page 5) as followed:

As predicted, long-term operation stability can also reappear in wider NO_3^- concentration of 0.1 M with slightly higher NH_3 yield (Supplementary Fig. 50-51), and intact morphology of MBM interface can be maintained after long time application at 1000 mA cm^{-2} (Supplementary Fig. 52)

In the meantime, the long-term WD stability of the fabricated bipolar membrane was also tested in the manuscript. The stability was tested in neutral (0.5 M Na_2SO_4) for 550 hours and the cross-section view of bipolar membranes were also detected. The Figure was also placed here to prove the same issue (Figure R7).

Figure R7. The interfacial structure of **a**, FBM (bipolar membrane with flat interface) and **b**, MBM after WD stability measurements.

Comment 6:

The stability provided by the authors is very good. But the electrode area of 1 cm^2 is too small for industrialization. The authors should do the experiment with larger electrode area.

Reply 6:

We quietly appreciate for the valuable advice and try our best to follow in the revised manuscript. In the present stage of manuscript, **all the experiment we re-conducted with the cathodic and anodic electrodes of 3 cm^2 (length of side: 1.73 cm)**, as shown in the Figure R8.

Figure R8. The enlarged cathodic and anodic electrodes prepared for NH_3 electroynthesis in the revised manuscript.

Here we might claim that 3 cm^2 is the largest active area that we are able to adopt currently, limited by the test equipment in lab. A further higher of active area requires new design of flow channel and pipeline, when more generated of gas transfer becomes rate-limiting step. New materials, experimental systems and flow cells with 25 cm^2 active area are in preparation, which can be expected to appear in our subsequent work. **According to the updated result, we didn't find any adverse effect brought by the area multiplying (3 folds of last electrodes)**, which can be partly demonstrated by the I-V curve of NH_3 electroynthesis with the established flow cell (Figure R9) as well as the systematic stability involved before (Figure R2).

Figure R9. I-V curves of NH_3 electrosynthesis with catalytic electrodes of 3 cm^2

We hope the supplementary evidence can further prove the NH_3 system as a potential candidate for industrial applications in the future. We revised the expression in the Method part as followed (page 19-20):

A flow cell for NH_3 electrosynthesis was composed of endplates, current collectors, flow fields, gaskets, electrodes and bipolar membranes, with electrolytes circulation powered by a double channel peristaltic pump as shown in Figure 5a and Supplementary Figs. 28-29. Co 3D nanoarray and the as prepared NiFe foam⁴⁸ were adopted to be cathodic and anodic catalysts. The effective working area for both anode and cathode electrodes is 3.0 cm^2 respectively, sandwiching a bipolar membrane a slightly larger to avoid electrolytes convection of both sides.

Comment 7:

Please explain in detail the specific principle and mechanism of double electrode membrane used in nitrate reduction, which should include each step of reaction.

Reply 7:

As your this suggestion, we rearrange and supplement the description of the necessity of using bipolar membranes (double electrode membrane that mentioned) to complete a continuous nitrate reduction. Here please let me first extract a part of Figure 1 in the manuscript (noted as Figure R10) for explaining the principle and mechanism. We would also like to make a schematic illustration of the general chemical structures of the three kinds of membranes in Figure R11 to explain more clearly.

Figure R10. Mechanism of using a bipolar membrane for nitrate reduction. i) and iii) depicted the ionic moving situation inner membrane when using an anion exchange membrane or cation exchange membrane for the same process, respectively. The direction of electric field applied is illustrated in the schematic figure.

Figure R11. General chemical structure of **a**, anion exchange membrane; **b**, bipolar membrane; **c**, cation exchange membrane.

All the followed explanation is based on the nitrite reduction **in alkaline system** (e.g. KOH). The figure R10 showed three different circumstances of using i) an anion exchange or ii) a bipolar membrane or iii) a cation exchange membrane as the separator for the same NH_3 electrosynthesis process.

To be best of our knowledge, the cation exchange membranes (usually Nafion series) are most commonly used for nitrate reduction (no matter in flow cell or H-cell). As shown in the Figure R11c, a there are a bunch of negative charged functional groups grafted on backbones of CEM (usually $-\text{SO}_3^-$), and they will form a Donnon exclusion for ions with same charge and selectively transport the cations in the electrolytes. As a result, even though this membrane can resist the mass crossover between catholyte and anolyte for a short period of time; however, only K^+ can act as the charge carrier to contribute to the current (iii of Figure R10). **After long-term operation, most K^+ will concentrate to cathode and a gradually increased voltage will be consumed to overcome the chemical potential gradient of K^+ .** So theoretically the cation exchange membrane cannot afford long period of nitrate reduction in alkaline system.

An anion exchange membrane was also adopted for the same process, for which we may also explain the drawbacks. As shown in the Figure R11 a, there are positively charged functional groups dispersed on backbones of an anion exchange membrane (mostly are quaternized ammonium groups), and similarly these groups will offer the AEM with Donnon repulsion to cations in the electrolyte. However, the selectivity of charged groups cannot distinguish the difference of ions with same charge, especially for the monovalent ions. **Therefore, NO_3^- accompany with OH^- to act as the current carrier and transfer across the membrane if the flow cell is equipped with AEL.** As a kind of reactant as well as a common contaminant, the diffusion of NO_3^- might not be acceptable in real applications.

As shown in the Figure R10 and R11, the bipolar membrane is composited with an anion exchange layer and a cation exchange layer, thus none of the ions in the electrolyte can cross the whole membrane and reach to the other side. In this situation, a bipolar membrane can share a function of water dissociation at the interface of two layers, namely, to produce the H^+ and

OH⁻ from H₂O and moving outwardly. **So that ionic balance of both sides can be kept with the help of bipolar membrane, which is anticipated with a continuous mode of operation.**

Following the valuable advice from the reviewer, we share the three parts of reactions that happened inner flow cell here to clarify.

Cathode:

Anode:

Bipolar membrane interface:

The overall reaction is as followed:

According to the explanation, some revisions of the original manuscript are also presented here (page 2-3):

Nevertheless, a monopolar ion exchange membrane, either an anion exchange membrane (AEM) or a cation exchange membrane (CEM), can hardly fulfil the process and will lead to severe ionic (alkali metal ions or NO₃⁻) crossover, as shown in Figure 1. A bipolar membrane (BM) composed of an AEM and an CEM, by means of which ions with opposite charge can be repulsed by Donnan potential and H⁺/OH⁻ can be produced by water dissociation (WD) at the interlayer¹⁸⁻²² and move outwardly to constitute circuit, provides prospective answer to the dilemma theoretically (see detailed discussion in ii of Supplementary Note 1).

Comment 8:

The authors statement “a bipolar membrane flow reactor for continuous NH₃ electrosynthesis is realized at 1000 mA cm⁻² with Faradaic efficiency of 94.7%, state-of-art yield rate of 75.1 mg h⁻¹ cm⁻² and stable operation durability of 100 hours.” However, in the figure 5, the FE may be low than 90% after 30 h. Therefore, the author's statement is not rigorous enough and should be revised.

Reply 8:

Thank you so much for the rigorous comment for the qualifier we adopted in the last version. **In the updated version of manuscript, all the advice is adopted to describe the result in a more accurate and appropriate way according to the new experimental data.** For the reason that we made substantial revisions of the bipolar membrane NH₃ electrosynthesis part of the original revision, most of the experimental data is updated and revised, and also the expressions and vocabulary are deliberately reconsidered as your suggestions to reach a better stage.

As you can see in Figure 5, we designed a new cathodic catalyst (so-called Co 3D nanoarray) for nitrate reduction and a relative high-level of NH_3 Faradaic efficiency as well as yield rate can still be maintained even though the NO_3^- concentration we adopted lowered to 2000 ppm. Moreover, as we stated in Reply 2, the bipolar membrane tested for the updated data was reinforced by an PTFE substrate, which to a large extent increase the operation stability during test.

The NH_3 Faradaic efficiency and yield rate we tested at 1000 mA cm^{-2} at new stage was grabbed as shown in the Figure R12.

Figure R12. **a**, NH_3 Faradaic efficiency and yield rate at different current density. **b**, The variation of Faradaic efficiency and yield rate during 100-hour stability test.

As we can find in the updated data, the NH_3 Faradaic efficiency at 1000 mA cm^{-2} of established system is 86.2 % and $68.4 \text{ mg cm}^{-2} \text{ h}^{-1}$, respectively. When it comes to the stability measurement, the **highest value** we measured for 100 hours is **92.5 %**, and the **lowest** one is **84.7 %**. Therefore, we believe the value separately measured “86.2” and “68.4” is reasonable to represent the property of established flow system. Based on this consideration, we revised the statement in the abstract, introduction, result and discussion and conclusion as followed:

(Abstract part, page 1) By solving inherent conflict of suppressing hydrogen evolution and enhancing water dissociation requirements at catalytic sites, a bipolar membrane, the established MBM flow reactor can realize a stable NH_3 electrosynthesis at 1000 mA over 100 hours cm^{-2} with Faradaic efficiency of 86.2 % and maximum yield rate of $68.4 \text{ mg h}^{-1} \text{ cm}^{-2}$ with merely 2000 ppm NO_3^- alkaline electrolyte. The results show huge potentials for artificial nitrogen cycling in the near future.

(Introduction, page 4) By coupling with Co 3D nanoarray cathode that both intrinsic activity and mass transfer intensified, continuous bipolar membrane NH₃ synthesis in flow reactor achieved at 1000 mA cm⁻² with Faradaic efficiency of over 86.2 % and high yield of 68.4 mg h⁻¹ cm⁻² using merely 2000 ppm NO₂⁻ alkaline electrolytes. Moreover, a >100 hours operation at 1000 mA cm⁻² also endorses the confident of using MBM in high efficiency and yield rate NH₃ synthesis technology from industrial effluents.

(Result and Discussion, page 14) Giving credit to MBM with ultrahigh WD rate and mass-transfer intensified cathodic catalyst, the NH₃ yield rate of established flow reactor increased linearly with current density and achieved a maximum NH₃ yield rate of 68.4 mg h⁻¹ cm⁻² (1000 mA cm⁻²) with an NH₃ FE of 86.2 % with 2000 ppm NO₃⁻ (Figure 5e), far exceeding most electrochemical strategies for NH₃ synthesis reported so far (see comparative data in Supplementary Table S2).

(Conclusion, page 17) By solving inherent conflict of suppressing hydrogen evolution and enhancing water dissociation requirements at catalytic sites, continuous NH₃ bipolar membrane electrosynthesis can be realized over 100 hours with merely 2000 ppm NO₃⁻, achieving NH₃ FE of 86.2 % and maximum yield rate of 68.4 mg h⁻¹ cm⁻² at 1000 mA cm⁻², which far exceeds most NH₃ synthesis process ever reported.

Comment 9:

It is not uncommon to realize large current at 1 M nitrate solution. Thus, some reports of high current density should also be summarized and supplemented

Reply 9:

Following your valuable suggestion, we add some data (e.g. *Adv. Sci.*2021,8, 2004, *Nature Nanotechnology* 17, 759–767, 2022) with >100 mA cm⁻² nitrate reduction in the following table (Table R1) and the whole data list can also be found in the Supplementary Table 2.

Table R1. Supplementary comparative data of nitrate reduction to ammonia with relatively high electrolytic current (> 100 mA cm⁻²) previously reported.

Electrochemical device	NH ₃ Faradaic efficiency/%	*Current density/ mA cm ⁻²	NH ₃ yield rate/ mg h ⁻¹ cm ⁻²	Catalyst
		batched H-cell	96	

	98	500 (stability)	~34	
flow-system H- cell	93	1000 (I-V test)	76.5	Ru-CuNW
	>90	400 (stability)	~29.6	
batched H-cell	90	~450	42.1 mg h ⁻¹ mg _{cat} ⁻²	Iron-Cyano Nanosheets
batched H-cell	89.6	194.31	2.89	Nano-Ag

Reviewer 2:

General Comments:

The manuscript explores the use of a bipolar membrane to synthesize ammonia from nitrate. The work relies on unrealistic conditions of treatment of high alkaline solutions containing high nitrate concentrations, which would fail to provide the solution of ammonia recovery outlined in the introduction. The manuscript contains many high-sounding but empty words, which results in a tedious lecture with very unclear message. The manuscript seems to be mostly focused on the study of membranes while using commercial electrodes that do not seem to be at the state of the art. Authors fail to identify key researchers on this hot-topic of N-economy and electrosynthesis of ammonia from nitrate. Indeed, language and description used falls far from the recommended in the field of ammonia electrosynthesis from nitrate. At some point it is quite unclear what is the research question or advance being proposed and mixes concepts of completely different electrified processes. The manuscript may be a better fit for a specialized engineering journals with focus on reactor design/behavior that report systematic studies. Schematics are unclear and do not depict the details or key aspects, some of them are even far from the experimental observations and extrapolate a subjective idea (do not describe observed characteristics of synthesized materials/membranes). Unfortunately, acceptance of the manuscript cannot be encouraged.

Reply:

We quietly appreciate for the time and effort the review kindly offered to review our manuscript of last version, which we think is helpful to promote the quality of this work. We also feel grateful for the comments the reviewer provided, for which we try our utmost to reach an acceptable answers/explanations/supplementary data. Because the comments were not shared point by point, we here prudently summarized into three aspects: **i) the absence of novelty in the applied catalysts; ii) the use of high concentration nitrate solutions; iii) the lack of professional expression or writing in the field of ammonia synthesis.** Following presents the reply to the comments of three points.

i) the absence of novelty in the applied catalysts

We hope it could be reclaimed that the novelty of our work is aiming at a electrosynthesis of ammonia in a continuous way (100 hours) under mild conditions by using bipolar membranes for the first time. We shall apologize that the significance and impact of adopting and designing a high performance bipolar in the present process is not stated in the clearest way, which might confuse the reviewer. Actually, we never deny that the hottest topic of N-economy and electrosynthesis of ammonia from nitrate is to design and fabricate a catalyst with high efficiency to convert low concentration NO_3^- into NH_3 . It is precisely based on the tremendous efforts of catalytic materials the previous researchers made that prompt us to reflect on the systematic optimization of NH_3 electrosynthesis devices. We find there is little research work of membrane selection and fabrication completed. **In fact, the cation exchange membranes (Nafion series) or the anion exchange membranes are most commonly used for nitrate**

reduction in the existed reported works (no matter in flow cell or H-cell); however, ionic balance cannot be maintained by these two kinds of membranes (which means K^+ or NO_3^- will crossover the membrane). Only a bipolar membrane can separate the ions by the Donnan potential of its functional layers, which can be referred to HCO_3^- electroreduction process (studied more with bipolar membrane, e.g. *ACS Energy Lett.* 2016, 1, 6, 1149–1153). The new problem is that commercial bipolar membrane cannot offer a large current density (*Nature Energy* 6, 339–348 (2021)), **so in this paper we mainly put forward a design for this new application and studied in detail.** We sincerely hope the description of our logic can serve the review for a better understanding of original intention and meaning of this work.

In addition, we also share anodic catalyst design in the updated version of this work. Following comments from the reviewers, we designed an all-Co 3D nanoarray cathodic catalyst to reach a higher performance than Commercial Co foam, **so that NH_3 electrosynthesis can be conducted in a lower concentration (2000 ppm) instead of 1 M.** The design principle is shown in Figure R13.

Figure R13. The design principle and the microscopic morphology of Co 3D nanoarray.

The overall thinking of the new catalyst design is to fulfill a lower concentration of nitrate reduction. As we noticed, severe HER will badly compete with NO_3^- RR in low NO_3^- concentration electrolytes if a high current density is pursued. Under this circumstance, mass (ionic) transfer plays the same important role as the intrinsic activity of the catalytic materials to make a better utilization of NO_3^- in the electrolytes. In term of the analysis, we i) select Co element that reported with high nitrate reduction activity and ii) construct thin nanosheets of Co to help to expose the active sites and iii) fabricate 3D mass transfer channel of Co metal framework. As shown in Figure R12, a multilayer Co structure was fabricated successfully, and more detailed information could be found in the new updated Supplementary Figs. 29-42.

We sincerely hope the detailed explanation of bipolar membrane adoption and the updated catalyst design can help to solve the concerns from the reviewer.

ii) the use of high concentration nitrate solutions

We thank the reviewer for the suggestions relevant to the concentration we adopted for nitrate reduction, which is professional and helpful for us to catch the mainstream point and promote the research work to a higher level. To follow this valuable advice, we to a large extent lowered the nitrate concentration we adopted (**from 1 M to 0.1 M or 2000 ppm**), which as far as we know is widespread in industrial effluents. As mentioned before, we specifically designed a Co 3D nanoarray cathodic catalyst, which realize a simultaneous enhancement of mass transfer and active sites. So that an acceptable performance can still be maintained in a low nitrate concentration.

Even though a 2000 ppm of nitrate solution was adopted, the NH_3 Faradaic efficiency at 1000 mA cm^{-2} of can still achieved 86.2 % and $68.4 \text{ mg cm}^{-2} \text{ h}^{-1}$, respectively, and the stability of NH_3 flow cell was promoted. All experimental data of bipolar membrane NH_3 electrosynthesis were re-collected with new materials and low nitrate concentrations, including I-V polarization curves, Faradaic efficiencies and stability test, as shown in Figure R14-16.

Figure R14. I-V curves of bipolar membrane electrosynthesis with a, 2000 ppm and b, 0.1 M nitrate.

Figure R15. NH_3 Faradaic efficiencies (lower bars), yield rates (upper circular scatter-line, left Y axis) and energy consumptions (upper rectangular scatter-line, right Y axis) of MBM NH_3 electrosynthesis system with a, 2000 ppm and b, 0.1 M NO_3^- .

Figure R16. Stability measurements of MBM NH_3 electrosynthesis systems with a, 1 M OH^- and 2000 ppm NO_3^- and b, 1 M OH^- and 0.1 M NO_3^- at 1 A cm^{-2} . The

lower plots show the NH_3 FE (circular scatter-line, left Y axis) and yield rates (rectangular scatter-line, right Y axis) versus time; the upper plot shows the cell voltage change versus time.

Moreover, the main part of descriptive expression of the NH_3 electrosynthesis was re-write according to the updated data. We heartily hope the concerns of high nitrate concentration from the reviewer could be released with the supplementary data.

iii) the lack of professional expression or writing in the field of ammonia synthesis.

We feel grateful for the strict comments on the expression and writing. We carefully go over the manuscript and supplementary materials to pick up all the inaccurate and unprofessional expressions related to NH_3 electrosynthesis as far as we could. Most of the wording, sentences and description have been re-thought over and polished, which was also yellow highlighted in the revised manuscript. We believe the renewed version achieved a better stage now for the request from the reviewer.

Reviewer 3:

General Comments:

The manuscript reported the design and use of bipolar ion exchange membrane for nitrate electroreduction. They claim that the highest current density can reach 1000 mA cm^{-2} , and the continuous ammonia production would exceed 100 hours. This is attractive data, but not novel. Similar achievements have been reported by other groups (See a recently published article: Adv. Sci., 2021, 8, 2004523. Metallic Co Nanoarray Catalyzes Selective NH_3 Production from Electrochemical Nitrate Reduction at Current Densities Exceeding 2 A cm^{-2}). In addition, the exploration of the mechanism and the design of the catalyst are mediocre. Therefore, this manuscript is not suitable for being published on Nature Communications.

Reply:

We unfeignedly appreciate for rigorous but useful advice from the reviews, which contributed a lot for us to promote the manuscript. Based on these pertinent comments, we made a large degree of revisions and data supplementary to the original manuscript, which will be listed point by point below.

We have already included the reference the reviewer mentioned in the comparative data and hope we could also share some discussions. The development of catalyst in many reported works have already realized an extremely high current density ($> 2 \text{ A cm}^{-2}$), especially with a relatively higher NO_3^- concentration. However, the 2.2 A cm^{-2} of NH_3 production was just a transient result based on the polarization curve, **and the stability tests were conducted under the relatively small currents (i.e., 200 or 500 mA cm^{-2}) for a short duration (i.e., 10 hours)**. Therefore, we considered a direct performance comparison might not be the best choice.

Moreover, we hope we could also reclaim the novelty and importance of fabricating and adopting a bipolar membrane for the nitrate reduction as a response to the concerns of the performance and catalyst innovation from the reviewer. We find that a bunch of catalysts have already achieved a Faradaic efficiency near 100 % with large yield rate, which made us believe catalytic materials should not be the limiting step in the near future and urged us to think about the **optimization of membrane selection and design** for application in the industrial device. We noticed that many reported works with high performance cannot afford a long-term stable operation, which can be account to the membrane selection. Here follows the explanation and takes alkaline electrolytes as an example. As stated, we find that a bipolar membrane that composited with an anion exchange layer and a cation exchange layer can simultaneously repulse the crossover of ions of opposite charge and maintain circuit by its water dissociation function, **so that ionic balance of both sides can be kept to achieve a sustainable nitrate reduction in flow cell**. Please find detailed explanation in the revised manuscript and reply 5.

In light of this, we would like to emphasize for the core idea: **i) the bipolar membrane system is feasible for a continue NH_3 synthesis from nitrate and ii) the bipolar membrane we designed overcome the commercial one to realize a higher current application.**

We also thank the reviewer for the comments in aspect of catalyst design. To better solve the worriers from the reviewer, we specifically design new cathodic catalysts and conducted particular characterizations on the materials, the performance of which far exceeds the Co foam we adopted before. **The primary target of designing a new catalyst is to simultaneously promise an intrinsic activity as well as promote the mass transfer on ions near the surface of catalysts, so that a lower concentration (2000 ppm) can be better utilized.** To achieve this goal, we selected Co (zero valance) as the element to construct nitrate reduction catalyst and built a multilayer structure by growing Co nanosheets on 3D Co framework (so-called Co 3D nanoarray) to extremely expose the activity sites of the catalyst and enhance the mass transfer process. The preparation, morphology and chemical structure could be referred in revised Supplementary Figs. 29-33. Some of the electrochemical data were presented below to prove the effectiveness of the catalyst.

Figure R17. The nitrite reduction performances of Co-based catalysts. **a**, I-V curves of fabricated Co catalysts. **b**, double layer capacitance of Co-based catalysts. **c**, NH_3 faradaic efficiency and yield of Co 3D nanoarray in 2000 ppm NO_3^- . **d**, NO_2^- (blue scatter-line) and N_2H_4 (yellow scatter-line) faradaic efficiency of Co 3D nanoarray catalyst.

According to Figure R17, Co 3D nanoarray can still afford an electrolytic current $> 1000 \text{ mA cm}^{-2}$, and the double layer capacitance of this structure also reveals as the highest one. For the reason that same Co element we adopted for each catalyst, we speculated that the unique structure of Co 3D framework can help to intensify the NO_3^- transfer the inner reaction sites with 2000 ppm electrolytes as our original design. Also, the NH_3 faradic efficiency was maintained in a higher current range with negligible side products detected (Figure R17c, d). The designed nitrite reduction catalyst was used for the bipolar membrane NH_3 electrosynthesis system, and all experimental data in this were re-conducted with merely 2000 ppm NO_3^- to avoid using the high concentration. More detailed characterizations for the designed catalysts could be found in updated Supplementary information (Supplementary Figs. 34-42 and Note 5).

We sincerely hope the substantial revision of the manuscript reach a better stage and could partly dispel the concerns from the reviewers.

Here we also present the description of catalyst design in revised manuscript for the convenience of review (page 13-14):

To achieve a high yield NH_3 electrosynthesis with relatively low NO_3^- concentration, we specifically designed a Co 3D nanoarray self-supported catalyst by electrodeposition, where Co nanoarray was densely aligned on Co framework (Figure 5c and Supplementary Figs. 29-32). To conquer the severe HER at high current, the principle of design is to enhance mass transfer by a multilevel structure, thus reactant NO_3^- can be easily transferred through 3D framework to Co nanoarray with high intrinsic reduction activity, which can also be proved by the depth of microporous and the bulky surface area (Supplementary Figs. 33-35, see detailed discussion in Supplementary Note 5). The unique structure of Co catalyst promised an over 90% NH_3 Faradaic efficiency (FE) with 2000 ppm NO_3^- as well as surpassed side products (NO_2^- and N_2H_4) at $>1000 \text{ mA cm}^{-2}$ (Supplementary Fig 36-42). Coupling with NiFe anode with low OER overpotential reported by us previously (Supplementary Figs. 43-44)⁴⁸, the MBM electrosynthesis system is prospective to work with high efficiency at ampere-level.

Comment 1:

Are Supplementary Fig 9e and 10e the same? There are many similar situations. Although the author may try to facilitate comparison, it will be very confusing to the readers.

Reply 1:

As suggested, we make further explanations and rearrange the figures in the Supplementary information. **We delete the Supplementary Figs. 9-12 in the last version and compact the contents. The revision can be referred in new Supplementary Figs. 9-10 in the updated version of Supplementary information.**

For a further description, Supplementary Figs. 9-10 present the surface and cross-section images of the templates, respectively. We put the SEM images of CoNi array template here to prove the following 2 points: i) A needlelike morphology could be maintained with whatever precursor concentrations or growth duration adopted; ii) The length and width of every needle could be regulated in sub-micrometers and micrometers. Locally amplified images were inset for all figures here. Similarly, we also rearrange the Supplementary Figs. 15-16 to show the surface morphology of template-transferred AELs using the CoNi arrays in Supplementary Figs. 9-10.

Comment 2:

Error bars should be added for important experimental data. In addition, the results are repeated for three times. Otherwise, the conclusion may be inaccurate.

Reply 2:

Following the suggestion, we conducted three parallel experiments for all updated data that relevant to NH₃ electrosynthesis part. **The error bars were added in the plots in updated version of manuscript and Supplementary information** (see Figure 5e, f in revised manuscript and Supplementary 46, 47, 49, 50 in SI):

Comment 3:

Why did the author choose NiCo LDH as the model catalyst? LSV measurements showed that the maximum current density was only 0.35A cm⁻² at -0.65V vs. RHE. This means that the actual applied voltage at -1A cm⁻² is more negative. A large number of research results show that the catalyst will undergo in-situ conversion at a very negative potential. Therefore, the stability of the catalyst after electrochemical test needs to be reconsidered. In addition, the long-term test results showed that the performance of the catalyst had declined. Is this caused by catalyst conversion or membrane changes?

Reply 3:

We genuinely appreciate the reviewer for these useful comments and are also willing to answer the doubts from the reviewers as far as we could. As we can understand, the comments could be divided into the following 4 points: **i)** The reason why we choose NiCo LDH. **ii)** The concerns of the low performance of nitrate reduction catalyst and the in-situ conversion situation we adopted. **iii)** The stability maintenance of the nitrate reduction catalyst. **iv)** The reason of the performance decline during NH₃ electrosynthesis. We could share our responses and present revisions point by point as followed:

i) The reason why we choose NiCo LDH

We shall apologies that it is not be expressed accurately enough for your understanding. Actually, **the CoNi hydroxide nanoarray was selected and adopted as a template to form**

the surface morphology of anion exchange layer and was dissolved and sacrificed in 6.0 M HCl in the bipolar membrane fabrication process. The reason why we choose to synthesis NiCo LDH as template is that the needle-like morphology of NiCo is appropriate to construct a “mortise-tenon” structure interlayer of bipolar membrane, and can be regulated in sub-micrometers via a simple hydrothermal method. Subsequent contents of the manuscript did not involve CoNi materials anymore.

On the other hand, the catalyst we adopted in the last version of manuscript for the nitrate reduction reaction is commercial pure Co foam without further modification, which is different from the NiCo. In the current version, we have already replaced the catalyst by a newly designed Co 3D nanoarray catalyst with much higher performance for nitrate reduction, which is discussed in the reply for general comment part.

To share a more accurate expression, we have also revised the relevant description as followed (page 5):

As shown in Figure 2b, the template was firstly prepared by self-growth of CoNi hydroxide on metal substrate under hydrothermal method to obtain a needle-like microarray, and then QPPT ionomer solutions was casted on this template surface for AEL fabrication (see chemical structure in Supplementary Figs. 7-8). After peeling off from the substrate and etching the embedded microarray, the AEL with micro-patterned surface was obtained. Finally, the fabrication of MBM was completed by spray-coating of WD catalysts ink and PFSA CEL on patterned side of AEL, hereto “mortise-tenon joint” structure was achieved.

ii) The concerns of the low performance of nitrate reduction catalyst and the in-situ conversion situation we adopted

We acknowledge that the performance of nitrate reduction catalyst is of a low level compared to the data ever reported, hampering the efficiency in a NO_3^- concentration catholyte. Following the advice from the review, **we designed and fabricated a Co 3D nanoarray catalyst for nitrate reduction and adopted it in the flow cell measurements.** The intention is to simultaneously combine the intrinsic activity of cobalt and mass transfer advantage, which has been discussed in detailed in the reply for general comment and in revised Supplementary Note 5. The electrochemical measurements were also optimized by mixing with a stir bar to mimic the flow in the cell. As can be found in Figure R17a, a 1000 mA cm^{-2} can be achieved. Thus, we believe the performance of newly designed catalyst for nitrate reduction should not be worried about.

We also approve the reviewer that in-situ conversion of catalyst might happen at a relatively negative potential, which is a common situation in previous reported works. With the performance promotion of new catalyst, a less negative potential should be applied to achieve 1000 mA cm^{-2} at current stage, which could alleviate the in-situ conversion situation. On the other hand, the fabrication process of designed Co 3D nanoarray contains three individual electrochemical processes, including a large-current electrodeposition, a chronoamperometry electrodeposition process and an electrochemical reduction process (Figure R18). In the last

step of fabrication, a quite negative potential (-1.4 V vs. Hg/HgO) was applied to the working electrode for a long period of time (20 minutes) in 1.0 M KOH, so that Co^{II} can be transformed into Co^0 . **We think the in-situ conversion of Co can be nearly completed during the fabrication process and little further conversion will appear in the following the NH_3 electrosynthesis.** The structural stability of Co is also proved by experimental data in the following reply iii).

Figure R18. Fabrication and detailed conditions for Co 3D nanoarray construction.

iii) The stability maintenance of the nitrate reduction catalyst

We realized the stability of catalyst for the nitrate and NH_3 electrosynthesis is a vital point to be considered. For figuring out this point, we conducted supplementary experiments to compare the chemical/morphology structure of catalyst before and after long-term NH_3 electrosynthesis measurement.

Figure R19. The stability measurements of cathodic catalyst. The morphology structure of Co 3D nanoarray before and after NH₃ electrosynthesis are showed in **a**, and **b**, respectively; **c**, the I-V curves of original Co catalyst and after used; **d**, the XRD pattern of catalyst before and after used.

As shown in Figure R19, the **morphology** of the Co 3D nanoarray with multi-structure kept ideally before and after over 100 hours of operation, which can prove the stability of constructed catalyst. Furthermore, **XRD pattern** also showed that plane (111) and (200) of Co maintained well during NH₃ synthesis. Ascribing to the stability of morphology and chemical structure, an acceptable maintenance of nitrate reduction performance can be obtained from **I-V curve** (Figure R19d).

We hope the added data can help to prove the stability of fabricated catalyst.

iv) The reason of the performance decline during NH₃ electrosynthesis

We also thank the reviewer for noticing the stability issue, which could be a key point in practical application. **We might firstly claim that the stability issue of the last version has been optimized and solved in 100-hour duration**, and the updated data has been presented in Figure 5f and Supplementary Figs. 50-51.

We reasonably speculated that the NH₃ Faradaic efficiency and yield rate decline in the last version can be caused by both the **Co foam decay and the product crossover**, and the later reason might bear main responsibility. The NH₃ diffused cross the bipolar membrane driving by concentration difference can lead to NH₃ reduction in catholyte, causing decrease of the calculated FE and yield, as well as increase of average energy consumption for per gram of NH₃. Another secondary evidence is that the cell voltage becomes lower after working for tens of hours, which can also be resulted by a part of NH₃ crossover from cathode and re-oxidized at the anode. Therefore, we try to attribute the root of the problem to the membrane layers, which enjoys low ionic resistance at the expense of dimensional stability in aqueous system.

Because the Co foam was replaced by Co 3D nanoarray in the latest manuscript, we do not conduct intensive research for the old one. **For solving NH₃ crossover through membrane, we added a PTFE substrate for the anion exchange layer to reinforce the membrane and control the swelling value.** The substrate we adopted is not only chemically stable, but also possess porosity over 90%, which result negligible influence for the original performance of MBM. The updated stability data showed that NH₃ faradaic efficiency and yield rate can be well maintained for over 100 hours (no matter with 0.1 M or 2000 ppm NO₃⁻).

For the revised experimental data, we updated the relevant expressions as followed (page 15):

Long-term stability of such reactor operated at high current density is vital for practical applications in industrial-scale artificial nitrogen cycle. For verifying, established reactor equipped with MBM was operated with 2000 ppm NO₃⁻ at a constant current 1000 mA cm⁻² mode for >100 hours, during which imperceptible cell voltage change can be observed (Figure

5f), in sharp contrast into one operated with Commercial BP1 (Supplementary Fig. 49). This disparity was coincident with the WD stability as discussed before and can also be attributed by “self-interlocked” effect of mortise-tenon joint. In the meantime, time-dependent products were also detected, revealing a maintenance of ~90% NH_3 FE ~70 $\text{mg cm}^{-2} \text{h}^{-1}$ yield rate as well as <1.5% side product NO_2^- over 100 hours working (Supplementary Fig 51). As predicted, long-term operation stability can also be achieved in wider NO_3^- concentration of 0.1 M with slightly higher NH_3 yield (Supplementary Figs. 50-51), and intact morphology of MBM interface can be maintained after long time application at 1000 mA cm^{-2} (Supplementary Fig. 52).

We hope the point-by-point response could appropriately solve the confusions and catch the advice well from the review.

Comment 4:

If the catalyst is converted in situ, a small amount of dissolved Ni or Co may interfere with the UV-vis results. Therefore, other quantitative methods of ammonia should be provided. NMR tests are recommended.

Reply 4

We sincerely appreciate for this professional advice, which we think is a potential source of error for NH_3 concentration determination. In order to avoid the inaccuracy of UV-Vis method interfered by the Co, Ni and Fe elements as well as other possible situation, **we selectively conducted an isotope labelling experiments under NMR measurements to verify the reliability of obtained NH_3 Faradaic efficiency.** The data verified was NH_3 Faradaic efficiency vs. current density with 2000 ppm NO_3^- reduction.

The experimental procedure we adopted are described as follow: Firstly, we conducted NH_3 electrosynthesis with same materials and conditions but replace the $^{14}\text{N-KNO}_3$ by 99% (atom) $^{15}\text{N-NO}_3$. **The goal of using isotope labelling N-source is to exclude other possible unknown interference of the system and insure all the obtained NH_3 was converted from $^{15}\text{N-NO}_3$.** 500 μL of electrolytes was taken out and neutralized to weak acid by 2 M HCl as sample and mixed with D_2O to achieve a total amount of 600 μL . The mixed electrolyte sample of ^{14}N and ^{15}N was qualitatively detected by ^1H NMR (Bruker, 400 MHz), and different peak splitting of H can be observed at previously reported (Figure R20a). Based on the difference, we could distinguish the “real” NH_3 product in the electrolytes after nitrate reduction. Secondly, a $^{15}\text{N-NH}_4\text{Cl}$ calibration line was obtained for further determination of NH_3 Faradaic efficiency. We prepared several concentrations of $^{15}\text{N-NH}_4\text{Cl}$ solutions (~90% H_2O & 10% D_2O , adding 2 M HCl till weak acid) with precise amount of maleic acid as external standard. The samples were detected under NMR and the concentration of ^{15}N could be indicated by the peak area ration between $^{15}\text{NH}_4^+$ and external standard, so that a calibration line can be obtained (Figure R20b). Thirdly, the NH_3 electrosynthesis at different current density (200, 400, 600, 800, 1000 mA cm^{-2}) were conducted with 2000 ppm $^{15}\text{N-KNO}_3$, and the NMR samples were prepared with same method. The concentration of NH_3 could be quantitatively determined according to the

calibration line, and the Faradaic efficiency could be calculated and compared with the UV-Vis method (Figure R20c).

Figure 20. N-15 isotope labelling experiments for NH_3 determination. **a**, ^1H NMR spectrum of products using $^{15}\text{NO}_3^-$ and $^{14}\text{NO}_3^-$ as N-source. **b**, calibration line of $^{15}\text{NH}_4^+$ concentration. **c**, the comparison of Faradaic efficiency between colorimetric method and isotope labelling method.

According to Figure R20c, the Faradaic efficiencies determined by isotope labelling experiments and UV-Vis method are close ($< 5\%$ error), indicating the reliability of the colorimetric method we adopted in this work. The existed tiny difference could be reasonably concluded to the systemic error of two methods.

We also supplement description and experimental data in the revised manuscript (Supplementary Fig. 47) and hope the result could clear the air about NH_3 interfere for the readers.

Comment 5:

The highlight of this paper is the selection of bipolar membranes to promote the migration of anions and anions. Therefore, the pH of the solution in the anode and cathode cells after electrochemical testing is an important parameter, especially after high current density and long-term electrochemical testing. The authors should mention these details.

Reply 5:

We quietly appreciate for the reviewer for paying attention to the highlight of this work and providing valuable advice. As shared before, the main goal that we adopted a bipolar membrane for the NH₃ electrosynthesis flow cell is to keep the ionic balance of both anode and cathode side, as shown in Figure 1 and Figure R21, which is achieved by the Donnon exclusion of membrane layers.

Figure R21. illustration schematic of reactions happening at both sides and ionic transportation during NH₃ electrosynthesis with an anion exchange membrane, cation exchange membrane and bipolar membrane.

Here we believe an analysis of anode electrolyte (OER happening) can share a helpful answer. Here we would taking the **anodic electrolyte** as an example. As shown, a single layer of anion exchange membrane can hardly resist the crossover of nitrate, so only a part of hydroxides produced at the cathode can migrate to the anode. As a result, the hydroxides consumed at the anode could only be partially replenished and the pH might gradually decrease. On the other side, a single layer of cation exchange layer will selectively transport K^+ , and hydroxides would not be replenished and leads to a higher decrease of pH value. **The bipolar membrane is supposed to maintain the ionic balance by its charge exclusion and water dissociation ability to fully replenish the hydroxides consumed at the anode**, so the pH value should not undergo an obvious change during electrosynthesis as an ideal situation.

For better proving this point in an experimental way, we measured the hydroxide content of the anodic electrolyte during the updated 100-hour stability measurements with new test conditions (2000 ppm NO_3^- , 1000 mA cm^{-2}). Because we found that water migration crossover the membrane can happen during long-term and high-current electrolysis (a common situation in electrochemical device) and cause a change of total volume, **so we take total mole number**

change of hydroxides in the anolyte as an indicator. Meanwhile, we choose titration method (phenolphthalein) instead of pH value measurements achieve a higher accuracy to answer for the reviewer. The hydroxide molar values were presented in Figure R22 as followed:

Figure R22. The change of hydroxide amount in anolyte vs. NH_3 electroynthesis time.

It could be found in the Figure R22, **the hydroxide amount was kept ideally during 100-hour NH_3 electroynthesis.** A slight decrease compared to the original value could be ascribed to the co-ion leakage, a common phenomenon of a bipolar membrane.

To prove the same point in a more comprehensive way, we also conducted an individual modelling experiment to figure out the ionic balance properties of three kinds of ion exchange membranes. As shown in the Figure R23, a 5000-second electrolysis was operated at 1000 mA cm^{-2} and the K^+/NO_3^- concentrations of anolyte was detected. Here K^+ was measured using inductively coupled plasma-mass spectrometry (ICP-MS) and NO_3^- was detected by colorimetric method.

Figure R23. The change of K^+ and NO_3^- concentration in anode electrolyte after 5000s NH_3 electroynthesis with flow cell.

As can be found, **only the bipolar membrane (MBM) can maintain both the cation and anion concentrations of each side during electroynthesis.** Based on the supplementary data, we revised the manuscript for a better understanding of the reader as followed (page 13):

For demonstrating necessity of using a BM in the established system, three kinds of ion exchange membranes were elected for comparison, in which circumstance only a BM can steadily maintain a nearly constant cell voltage with 1000 mA cm^{-2} electrolytic current for 5000

seconds (5000 coulombs transferred), while the same mission can hardly be achieved by cation or anion exchange membranes (Figure 5b). Meanwhile, the anodic K^+ and NO_3^- concentration variation proved that merely BM can realize an ionic balance via WD, but oppositely severe ionic crossover occurred in CEM (K^+) or AEM (NO_3^-) cases, leading to instability of this process.

We expect the supplied data and evidence can achieve a better explanation and significance of using a bipolar membrane, so that the suggestions from the reviewer could be kindly replied.

Comment 6:

The authors are encouraged to provide the original data.

Reply 6:

Following the guidance and the requirements to the author for publishing in Nature Communications, we have well prepared all the source data for the Figures referred in the updated manuscript and Supplementary information in two separate Excel documents. The data can be uploaded immediately as long as the manuscript comes to a potentially acceptable state as requested by the editor. Moreover, the original data in other file formats is also available from us if necessary.

Comment 7:

To facilitate researchers to compare these results, ECSA normalized data should be provided to exhibit the intrinsic activity of this catalyst. The author summarize the recent reports so that the author and readers can understand the progress of nitrate electroreduction.

Reply 7:

We thank for the advice and inferred that the catalyst the reviewer mentioned here should be referred to the nitrate reduction catalyst or OER catalyst that adopted for the cell rather than a membrane relevant issue. (As in the Reply 3, the CoNi was only adopted as a template for the construction of bipolar membrane interface instead of one catalyst, so no further electrochemical properties were investigated for the template material.)

In the updated version, we designed new nitrate reduction catalysts to replace the Co foam for a higher performance of NH_3 electrosynthesis. Following the suggestion as we could, we conducted cyclic voltammetry for the nitrate reduction catalysts and figured out the ECSA for each catalyst, and then **we plotted the ECSA-normalized LSV data in Figure R24 (see CV plots in Supplementary Fig. 34).**

Figure R24. **a**, ECSA values of constructed catalysts for nitrate reduction; **b**, ECSA-normalized LSV data of cathodic catalysts.

We noticed that Co 3D nanoarray with the highest performance in 2000 ppm NO_3^- electrolytes enjoys the highest ECSA compared to others; oppositely, this catalyst can only exhibit the lowest ECSA normalized current density at the same potential (2.2 mA cm^{-2} at 0.2 V). It can be simply concluded that the multilayer structure of Co 3D Nanoarray didn't reveal intrinsic activity advantage compared to other Co materials, and also further proved that mass transfer step might be the limiting step for Co-based catalyst in a relatively low NO_3^- concentrations.

We have adopted the NiFe catalyst that we reported previously (*ACS Appl. Energy Mater.* 2021, 4, 9, 9022–9031) for OER in the flow cell, which possess easy fabrication process, acceptable performance, and stability. We also supplement ECSA related data here for the advice from the reviewer as in the Figure R25 (Supplementary Fig. 43).

Figure R25. **a**, Plots of current density versus the scan rate for various catalysts for ECSA measurements; **b**, ECSA-normalized LSV data of OER catalyst.

We also thank the reviewer for the suggestions on summarizing recently reported nitrate reduction work. As a complementary to the comparison data (Supplementary Table 2), we also added several research works with outstanding NH_3 yield and high electrolytic currents into the original one (e.g. *Adv. Sci.* 2021, 8, 2004, *Nature Nanotechnology* 17, 759–767, 2022). We also listed the supplementary part in Table R2 as followed:

Table R2. Supplementary comparative data of nitrate reduction to ammonia with relatively high electrolytic current.

Electrochemical I device	NH ₃ Faradaic efficiency/%	*Current density/ mA cm ⁻²	NH ₃ yield rate/ mg h ⁻¹ cm ⁻²	Catalyst
batched H-cell	96	2200 (I-V test)	176.8	Metallic Co
	98	500 (stability)	~34	Nanoarray
flow-system H- cell	93	1000 (I-V test)	76.5	Ru-CuNW
	>90	400 (stability)	~29.6	
batched H-cell	90	~450	42.1 mg h ⁻¹ mg _{cat} ⁻²	Iron-Cyano Nanosheets
batched H-cell	89.6	194.31	2.89	Nano-Ag

We hope the revised contents and updated data could appropriate response this comment from the reviewer.

Comment 8:

Some grammatical errors should be double-checked.

Reply 8:

We carefully go over the manuscript and supplementary materials to pick up all the inaccurate and unprofessional expressions. The new version has also been polished with the help of an

English native speaker. We consider the up-to-date version can achieve a better level to meet the high requirements of the professional editor and reviewer.

REVIEWER COMMENTS

Reviewer #1 (Remarks to the Author):

The authors carefully revised the manuscript and addressed the questions. The article has been improved, and it is recommended for publication.

Reviewer #3 (Remarks to the Author):

The author described a bipolar membrane with a mortise-tenon joint interfacial structure for nitrate-to-ammonia conversion. This design can effectively promote hydrolysis and inhibit hydrogen evolution, thus achieving a long-term stable nitrate electroreduction process.

The authors have made a lot of modifications, rewrote the manuscript, and answered my concerns. Now, this work highlights the design of the bipolar membrane structure to improve the mass transfer resistance of the cathode and anode.

At present, ampere-level current density has been reported, including the key parameter of realizing 1 A current density, more than 80% FE, and high stability tests. Thus, the catalytic novelty is not attractive, and bipolar membrane has well explored for various reactions from CO₂ reduction to H₂O₂ electrosynthesis, but the manuscript is interesting for scientists in the chemical engineering of nitrate reduction.

Some new comments:

1. The statement that bipolar membrane inhibits HER needs to be reconsidered. Because the design of the ion exchange membrane is difficult to affect the selectivity of the catalyst, they are two different parts in the electrolytic cell. The reason for the inhibition of HER should be determined by the adsorption energy of active hydrogen on the catalyst surface and the transfer capacity of active hydrogen.
2. The intrinsic activity is the activity normalized by the electrochemical active area. In fact, the intrinsic activity of self-supporting Co is not high, and its advantage lies in the high current density with geometric area normalized.
3. The alternative to the H-B process is inappropriate. The significance of the H-B process lies in nitrogen fixation and ammonia synthesis. At present, the preparation of nitrate in the world is based on the H-B process and subsequent Ostwald process. The authors need to revise the related description.
4. The overpotential for nitrate reduction at large current densities is significant. It is suggested that the authors compare their overpotentials with the recent reports.

5. The authors demonstrate the new applications of a bipolar membrane in nitrate electroreduction. A techno-economic analysis of their bipolar membrane will further improve the quality of this work.

So, I would like to recommend this work for being published after addressing my new concerns.

Response to reviewers

Manuscript NCOMMS-22-31417A-Z

We quietly feel grateful for the editor for further consideration of our manuscript and also highly appreciate for a lot of time and efforts paid from all reviewers. We are thankful for the positive comments as well as other constructive suggestions, which we believe are helpful for us to promote the manuscript in a multi-dimension way and get to a better stage of this work. Now all the comments were carefully considered and the manuscript was properly revised. The point-by-point responses are appeared as below.

Reviewer 1:

General comments:

The authors carefully revised the manuscript and addressed the questions. The article has been improved, and it is recommended for publication.

Reply:

We highly appreciate for the acceptance from the reviewer.

Reviewer 3:

General comments:

The author described a bipolar membrane with a mortise-tenon joint interfacial structure for nitrate-to-ammonia conversion. This design can effectively promote hydrolysis and inhibit hydrogen evolution, thus achieving a long-term stable nitrate electroreduction process.

The authors have made a lot of modifications, rewrote the manuscript, and answered my concerns. Now, this work highlights the design of the bipolar membrane structure to improve the mass transfer resistance of the cathode and anode.

At present, ampere-level current density has been reported, including the key parameter of realizing 1 A current density, more than 80% FE, and high stability tests. Thus, the catalytic novelty is not attractive, and bipolar membrane has well explored for various reactions from CO₂ reduction to H₂O₂ electrosynthesis, but the manuscript is interesting for scientists in the chemical engineering of nitrate reduction.

Reply:

The positive comments with tolerance for some inaccuracies and deficiencies in the old version of the manuscript are appreciated by the authors. We are also thankful for the comprehensive summary of the manuscript and try our best to offer accurate reply and appropriate revisions for the manuscript based on the 5 new valuable suggestions from the reviewer.

Comment 1#:

The statement that bipolar membrane inhibits HER needs to be reconsidered. Because the design of the ion exchange membrane is difficult to affect the selectivity of the catalyst, they are two different parts in the electrolytic cell. The reason for the inhibition of HER should be determined by the adsorption energy of active hydrogen on the catalyst surface and the transfer capacity of active hydrogen.

Reply 1#:

We quietly appreciate for the reviewer for offering this professional advice on the consideration and expressions appeared in the manuscript, and we also totally agree with this comment that the selectivity of main/side reactions are only decided by the surface situation of catalysts. The bipolar membrane that we designed and adopted here **can primarily help to balance ion transportation** and realize a long-term operation, as we try to state in the manuscript and last version of the response document (shown in Figure R1).

Figure R1. Schematic illustration of reactions and ionic transferring in bipolar membrane for NH_3 electrosynthesis process.

What the authors hope to express about the influence for the reactions lead by a bipolar membrane at the beginning is **its adaptivity in an alkaline system**. We thought a **higher pH with low concentration of H^+ can effectively suppress HER** and enhance the selectivity. Following the valuable advice from the reviewer, we found the brief but inaccurate description might lead to a misunderstanding of the readers, and we carefully examine the relevant expressions in the whole manuscript and revised/rewrote them.

Please see the corresponding revisions in the revised manuscript:

(Abstract, page 1-2)

By simultaneously boosting ionic transfer and creating bulky water dissociation sites for a bipolar membrane, the established MBM flow reactor can realize a stable NH_3 electrosynthesis at 1000 mA over 100 hours cm^{-2} with Faradaic efficiency of 86.2% and maximum yield rate of $68.4 \text{ mg h}^{-1} \text{ cm}^{-2}$ with merely 2000 ppm NO_3^- alkaline electrolyte.

(Conclusion, page 17)

With the adoption of bipolar membrane with fast kinetics and ampere-level water dissociation ability, continuous NH_3 bipolar membrane electrosynthesis can be realized over 100 hours with merely 2000 ppm NO_3^- , achieving NH_3 FE of 86.2% and maximum yield rate of $68.4 \text{ mg h}^{-1} \text{ cm}^{-2}$ at 1000 mA cm^{-2} , which far exceeds most NH_3 synthesis process ever reported.

Comment 2#:

The intrinsic activity is the activity normalized by the electrochemical active area. In fact, the intrinsic activity of self-supporting Co is not high, and its advantage lies in the high current density with geometric area normalized.

Reply 2#:

We are thankful for the reviewer for this useful comment, which prudentially reminded us on the catalyst-related words we used. As indicated by the reviewer, the relative high current of NH_3 electrosynthesis with low concentration NO_3^- was achieved by the 3D structure of the self-supported catalyst. Due to this micro-morphology, **the catalyst can bring about more effective active sites and get to better utilization of the NO_3^-** . To the best of our knowledge, the element Co lies in the first tier to catalyze the $8e^-$ nitrate reduction reaction (*Nature Energy* volume 5, pages 605–613 (2020)) but might represent disadvantages on its intrinsic activity compared to some reported ones (e.g. Ru). As a comprehensive consideration of cost and activity, we finally select Co to construct the 3D catalytic framework.

Apart from the discussion above as well as experimental data we obtained, we also presented the polarization curves and Tafel plots here to further explain the reason why we approve this comment from the reviewer (Figure R2).

Figure R2. Comparison of I-V relationships for Co catalysts with different morphologies: **a**, 2000 ppm KNO_3 ; **b**, 0.1 M KNO_3 . **c**, and **d**, show the Tafel plots, respectively.

As demonstrated in the Figure R2, even though the catalytic activity of 4 kinds of Co based cathode revealed huge difference at high current, **the Tafel slope of them was close to each other**. This phenomenon repeatedly indicate that the catalysts should possess **similar intrinsic activity** if the element, valence state and electronic structure are all the same.

Based on all the discussion, we would like to reasonably follow the suggestion from the review and revised the expression referred in the manuscript as followed:

(Abstract, page 1)

The design of MBM anticipate a continuous and high current NH_3 electrosynthesis from NO_3^- , and a Co 3D nanoarray cathode with large catalytic area and intensified mass transfer was fabricated by us to fulfill the goal.

(main text, page 4)

By coupling with Co 3D nanoarray cathode that both catalytic sites and mass transfer boosted, continuous bipolar membrane NH_3 synthesis in flow reactor achieved at 1000 mA cm^{-2} with Faradaic efficiency of over 86.2% and high yield of $68.4 \text{ mg h}^{-1} \text{ cm}^{-2}$ using merely 2000 ppm NO_3^- alkaline electrolytes.

(main text, page 13)

To conquer the severe HER at high current, the principle of design is to enhance mass transfer by a multilevel structure, thus reactant NO_3^- can be easily transferred through 3D framework to Co nanoarray possessing abundant catalytic sites, which can also be proved by the depth of microporous and the bulky surface area (Supplementary Figs. 33-35, see detailed discussion in Supplementary Note 5).

(conclusion, page 17)

The design of MBM anticipates a continuous and high current NH_3 electrosynthesis from NO_3^- , and a Co 3D nanoarray cathode with high abundant catalytic sites and intensified mass transfer was specifically fabricated to fulfil the goal.

Comment 3#:

The alternative to the H-B process is inappropriate. The significance of the H-B process lies in nitrogen fixation and ammonia synthesis. At present, the preparation of nitrate in the world is based on the H-B process and subsequent Ostwald process. The authors need to revise the related description.

Reply 3#:

We highly appreciate for this valuable advice from the reviewer and totally agree with the comment. We rewrote the description with a good grace according to the suggestion and try to emphasize the put forwarded process **as an emerging technology to fix the nitrogen in the waste industrial effluents** and play as a supplement role for the existed H-B process for NH_3 production.

Please see the revision presented below:

(conclusion, page 17)

Therefore, the continuous bipolar membrane NH_3 electrosynthesis with MBM was expected to alleviate nitrate contamination issues and partially compensate for huge global NH_3 consumption

with normal temperature and pressure process in the future.

Comment 4#:

The overpotential for nitrate reduction at large current densities is significant. It is suggested that the authors compare their overpotentials with the recent reports.

Reply 4#:

We thank the reviewer very much for reminding us to compare the overpotential of proposed process, and we would like to provide a detailed comparison list as elaborate as we could.

Before we compare the data, we hope we can make two simple statements for the performance comparison going as followed: i) Most advanced research works reported recently mainly focused on the catalytic materials design instead of membranes/overall systems, so only the potentials vs. RHE (obtained via three-electrode measurements) were presented. **Hence, we merely compare the catalytic potential (versus RHE) among reported catalysts and Co based catalysts in this work,** and the advantage of bipolar membrane design might not be easily revealed through the comparison. ii) The systematic potential (measured via two-electrode measurements) of the whole flow cell could also be obtained, which jointly contributed by the cathode and anode catalyst and the resistance of the bipolar membrane. **For cell potentials were seldomly reported, the overpotential of the MBM cell was only compared with the cell using commercial bipolar membrane.**

Figure R3. Comparison data of I-V relationships of NO₃⁻RR catalysts that can offer current > 100 mA cm⁻² recently reported: current density versus potential (RHE).

Figure R4. Relationship between current density and cell overpotential of the bipolar membrane NH_3 electro-synthesis flow cell (2000 ppm NO_3^-).

As demonstrated in Figure R3, most cathodic catalysts designed for $8e^- \text{NO}_3^-$ RR that can offer high current (100 mA cm^{-2}) were collected and compared with the self-supported Co 3D nanoarray catalyst. The thermodynamic potential for NO_3^- RR half reaction in alkaline conditions was 0.69 V (*Nature Communications*, 2021,12, 2870). As shown, the Co based catalyst used for bipolar membrane cell consumes **relatively lower overpotential** compared to most referred catalysts, which possess an enough performance to prove the advancement and effectiveness of bipolar membrane electro-synthesis system. On the other hand, the NH_3 electro-synthesis current density versus cell overpotential was also replotted in Figure R4 to achieve a better reply for the reviewer. In this figure, NH_3 flow cell equipped with different bipolar membranes were included and the significance of membrane developing was easy to be found. Worth noting that the thermodynamic voltage of the overall reaction (1.2 V) was deducted in the plot, so that **cell overpotential includes overpotential of electrode reactions (NO_3^- RR, OER) and ionic transferring resistance in the membrane.**

For further description, the main text was revised as followed (page 14):

By deducting the thermodynamic voltage for the reaction, MBM consumed much lower overpotential on the generating and transporting of ions compared to BP1 (Supplementary Fig. 46).

The Figure R3 & 4 were also added into Supplementary information as Supplementary Fig. 38 and Supplementary Fig. 46. The corresponding legends and numbers in the manuscript were revised.

Comment 5#:

The authors demonstrate the new applications of a bipolar membrane in nitrate electroreduction. A techno-economic analysis of their bipolar membrane will further improve the quality of this work.

Reply 5#:

The advice to supply a techno-economic analysis for membrane materials, device and the proposed process from the review is high appreciated. According to this, several related calculations were conducted to simply evaluate the cost of membrane fabrication and the process as followed.

i) The calculation of MBM fabrication cost and comparison with commercial bipolar membranes.

Table R1. Chemical prices referred in the bipolar membrane fabrication process. The prices of product can be searched on <https://www.alibaba.com>.

Name of chemicals	Price (P, \$ kg ⁻¹)
p-Terphenyl	219.7
1-Methyl-4-piperidone	77.0
Trifluoromethanesulfonic acid	68.6
Trifluoroacetic acid	29.9
Iodomethane	156.7
CoCl ₂ ·6H ₂ O	23.3
NiCl ₂ ·6H ₂ O	29.6
Ni plate	84.3
PFSA solution	195.5
Dichloromethane, Urea, etc.	18.0
Total	517.5

The cost of chemicals and consumable items for producing MBM can be calculated:

$$COST_{MBM} = \sum Usage\ amount \times P = \$517.5\ m^{-2}$$

According to the data from DONGYUECHEM Co., the total utilities payment takes up 10-15% of consumable items. Here we set 15% as an example:

$$Total\ COST_{MBM} = 150\% \times COST_{MBM} = \$595.1\ m^{-2}$$

Table R2. The costs comparison of MBM and other commercial bipolar membranes. The data can be obtained from fuelcellstore.com or <https://www.alibaba.com>.

Name	MBM	Fumasep FBM	TRJBM	Xion-BPM*	Neosepta BP1
------	-----	----------------	-------	-----------	-----------------

Price (\$ m ⁻²)	595.1	4733.3	2182.8	49671.1	1350
-------	--------	--------	---------	------

*Only small size is available. The data was calculated from the price of small size membrane

As shown in the Table 2, **the cost of MBM represent advantage compared to several commercial bipolar membranes.** To be mentioned here, the price of chemical and consumable items taken for calculation here is grabbed as lab-scale price, and might be further decreased in industrial production. Moreover, the PFSA cation exchange layer could also be reconsidered and replaced by other cationic materials with similar function and much lower price, which can obviously squeeze the overall producing cost in further applications.

ii) The techno-economic analysis of bipolar membrane NH₃ electrosynthesis process.

The analysis of techno-economic analysis for NH₃ electrosynthesis is based on the operation of 1 A cm⁻² and NH₃ FE of 90%. Assuming the flow rate in the electrolysis system as 0.1 mL/min in single pass mode, the daily production was assumed to be 100 ton/day. According to this condition, total current was calculated to be 58.4 MA, and the total area was 5839 m² with a total power usage of 174 MW. Based on these parameters, several levelized costs were shown in Table R3.

Table R3. Levelized cost for techno-economic analysis

Name of parameter	Value
Per area cost*	\$3576/m ²
Total cost of flow system	\$20.8 million
BoP cost**	\$11.2 million
Electricity cost***	\$125280/day
Maintenance cost****	\$3728/day

* Per area cost of the stack is estimated for 1 A/cm² operation condition. Assuming the instillation factor is 1.2.

** From the DOE analysis, the balance of plant capital cost is 35% of the total cost.

*** The price of electricity was assumed as \$0.03 /kWh.

**** The maintenance cost is assumed as 2.5% of capital cost per year.

Assuming the price of product is \$1536 /ton, the yearly profit is given as followed:

$$\begin{aligned}
 & \text{yearly profit} \\
 & = \text{the yearly income of products} - \text{yearly electricity cost} \\
 & \quad - \text{yerly maintenance cost} \\
 & = \$8.9 \text{ million /year}
 \end{aligned}$$

Assuming an ideal situation for the stack to be maintained, the NH₃ stack is designed for a 5-year of working. The NPV is roughly estimated using the yearly profit value as cash flows per year, and the nominal interest rate is 5%. According to the NPV calculation equation:

$$NPV = \sum_{n=1}^5 \$8.9 \text{ million} \times \left(\frac{1}{1.05^n}\right) - \$20.8 \text{ million} - \$11.2 \text{ million} = \$15.43 \text{ million}$$

The NPV of bipolar membrane electrosynthesis could be determined by the project life span, which was revealed as followed:

Figure R5. The relationship between calculated NPV and bipolar membrane NH₃ electrosynthesis device life span.

In conclusion, NPV becomes positive when the end-of-life span is around 4 years, which **demonstrates this project is a profitable investment.** To be mentioned here, most parameters referred here is obtained from lab-scale data, and the cost can definitely be further reduced for industrial-level application. Furthermore, the nitrate reduction is also an eco-friendly technology by solving contamination problem with renewable electricity, which can probably acquire preferential policy in the future. **The new technology also brings huge environmental benefits.**

According to the suggestions from the reviewer to further improve the manuscript, the above was also supplied as Note 6 in the Supplementary information doc. Relevant revision in the manuscript was highlighted as below (page 15):

An overall evaluation of material fabrication cost and process techno-economy (Supplementary Note. 6) also indicates 3D interfaces with mortise-tenon joint structure hopefully endows bipolar membrane with ability to withstand harsh operation conditions and delivers a promising pathway to accelerate the development of NH₃ electrosynthesis.

REVIEWERS' COMMENTS

Reviewer #3 (Remarks to the Author):

The authors have addressed my concerns. I do appreciate their efforts. Now, I believe that this is a well-organized manuscript with greatly improvement. Thus, I recommend it being published by Nat. Common.